# RealMAN: A Real-Recorded and Annotated Microphone Array Dataset for Dynamic Speech Enhancement and Localization

**Bing Yang[1,3], Changsheng Quan[1], Yabo Wang[1], Pengyu Wang[1], Yujie Yang[1],**
**Ying Fang[1], Nian Shao[1], Hui Bu[2], Xin Xu[2], Xiaofei Li[1,3]***

[1]School of Engineering, Westlake University
[2]Beijing AIShell Technology Co. Ltd
[3]Institute of Advanced Technology, Westlake Institute for Advanced Study
{yangbing, quanchangsheng, wangyabo, wangpengyu, yangyujie,
fangying, shaonian, lixiaofei}@westlake.edu.cn
buhui@aishelldata.com, xuxinwbg@163.com

## Abstract

The training of deep learning-based multichannel speech enhancement and source localization systems relies heavily on the simulation of room impulse response and multichannel diffuse noise, due to the lack of large-scale real-recorded datasets. However, the acoustic mismatch between simulated and real-world data could degrade the model performance when applying in real-world scenarios. To bridge this simulation-to-real gap, this paper presents a new relatively large-scale **Real**-recorded and annotated **M**icrophone **A**rray speech&**N**oise (**RealMAN**) dataset[2]. The proposed dataset is valuable in two aspects: 1) benchmarking speech enhancement and localization algorithms in real scenarios; 2) offering a substantial amount of real-world training data for potentially improving the performance of real-world applications. Specifically, a 32-channel array with high-fidelity microphones is used for recording. A loudspeaker is used for playing source speech signals (about 35 hours of Mandarin speech). A total of 83.7 hours of speech signals (about 48.3 hours for static speaker and 35.4 hours for moving speaker) are recorded in 32 different scenes, and 144.5 hours of background noise are recorded in 31 different scenes. Both speech and noise recording scenes cover various common indoor, outdoor, semi-outdoor and transportation environments, which enables the training of general-purpose speech enhancement and source localization networks. To obtain the task-specific annotations, speaker location is annotated with an omni-directional fisheye camera by automatically detecting the loudspeaker. The direct-path signal is set as the target clean speech for speech enhancement, which is obtained by filtering the source speech signal with an estimated direct-path propagation filter. Baseline experiments demonstrate that i) compared to using simulated data, the proposed dataset is indeed able to train better speech enhancement and source localization networks; ii) using various sub-arrays of the proposed 32-channel microphone array can successfully train variable-array networks that can be directly used to unseen arrays.

---

*Corresponding author.

[2]The RealMAN dataset is publicly available at `https://github.com/Audio-WestlakeU/RealMAN`.

# 1 Introduction

Microphone array-based multichannel speech enhancement and source localization are two important front-end audio signal processing tasks [1, 2, 3]. Currently, most existing works are deep learning-based and data-driven, for which the amount and diversity of microphone array data are crucial for the proper training of neural networks. Although an unlimited amount of microphone array data can be generated using simulated room impulse response (RIR) (with the image source method, ISM [4]) and multichannel diffuse noise [5], there are still significant mismatches between the acoustic properties of the simulated and real-world data, including **1) The directivity of microphones and sound sources, as well as the wall absorption coefficients.** ISM [4] assumes both microphones and sound sources to be omni-directional, while real microphones (especially when mounted on specific devices) have a certain directivity and human speakers have a frontal (on-axis) directivity [6]. In addition, ISM normally uses frequency-independent wall absorption coefficients, which violates the case of real walls. In [7], the authors introduced the frequency-dependent microphone directivity, sound source directivity and wall absorption coefficients into ISM, which largely improves the realness of simulated RIRs. **2) Room geometry and furniture.** ISM normally simulates empty shoebox-shaped rooms, while real rooms could have irregular geometry and built-in furniture. For simulating irregular rooms and built-in furniture, more advanced (but less computationally efficient) geometric acoustic (ray-tracing) methods can be used [8]. One extra difficulty is to configure the room geometry and layout, room and furniture materials to conform to the real-world scene semantics. In [8], this problem is resolved by leveraging scene computer aided design (CAD) models for room design and natural language processing techniques for material setup, which largely improves the realness of simulated RIRs as well. **3) Piece-wise simulation of moving sound source [9].** To simulate microphone signals from a moving sound source, the continuous trajectory of the source must be discretized into densely-distributed locations. One signal segment is simulated for each location, and signal segments are then connected. If the sound source moves too fast or the discretization is too sparse, the resulting microphone signals may contain clicking noises and other audible artifacts [9]. **4) The spatial correlation of simulated multi-channel noise** is normally determined under the hypothesis of a theoretical diffuse noise field [5], while the real-world ambient noise is a superposition of a large number of time-varying (partially) diffuse and directional noise sources and its spatial correlation could largely deviate from the theoretical values. Overall, due to the simulation-to-real mismatch, relevant studies on various tasks have shown that models trained with simulated data often perform poorly in real-world scenarios [7, 10, 11, 12].

Training with real-world data can avoid these mismatches. However, existing real-world microphone array data with annotated target clean speech and source location information is limited and lacks diversity. To address this, we collect a new **Real**-recorded and annotated **M**icrophone **A**rray speech&**N**oise (**RealMAN**) dataset from a variety of real-world indoor, outdoor, semi-outdoor, and transportation scenes. These recordings encompass diverse spatial/room acoustics and noise characteristics. A loudspeaker is used for playing source speech signal (although the used loudspeaker has a similar frontal on-axis directivity as human speakers [6], note that one fixed loudspeaker cannot account for the various directivities across different human speakers), and about 35 hours of clean Mandarin speech are used as source speech (more speech materials and for other languages will be considered in future recording if copyright can be issued). The dataset consists of 83.7 hours of speech and 144.5 hours of noise, recorded in 32 and 31 different scenes respectively, with both speech and noise recorded in 17 of these scenes. Both static and moving speech sources are included. We provide annotations of direct-path target clean speech, speech transcription, and source location (azimuth angle, elevation angle and distance relative to the microphone array), for speech enhancement (and evaluation of automatic speech recognition, ASR) and source localization. Where the direct-path target clean speech is obtained by filtering the source speech with estimated direct-path impulse responses, and the source locations are annotated with an omni-directional fisheye camera. A 32-channel microphone array is used for recording. End-to-end speech enhancement and source localization models are normally array-dependent, namely the network trained with one specific array can be only used for the same array. However, collecting real-world data for every new array is cumbersome. One solution is to use data from various arrays to train a variable-array network that can generalize to unseen arrays [13, 14, 15]. Our 32-channel array can provide many different sub-arrays for training such variable-array networks. Baseline experiments have been conducted on the proposed dataset, demonstrating that 1) Compared with using simulated data, training with the proposed real data eliminates the simulation-to-real problem and achieves better performances

in speech enhancement and source localization. Thus, the proposed dataset is more suitable for benchmarking new algorithms and evaluating their actual capabilities; 2) the variable-array networks [13, 15] can be successfully trained with our 32-channel array dataset. Hopefully, these networks can be applied directly to real applications involving unseen arrays.

## 2 Related Work

It is challenging to collect a large-scale real-recorded and annotated microphone array dataset. Table 1 and 2 summarize the existing multi-channel speech and noise datasets, respectively.

**Multi-channel speech recording and annotation.** In MIR [16], BUTReverb [17], Reverb [18], DCASE [19], ACE [20] and dEchorate [21], real-world RIRs are measured instead of directly collecting speech recordings. Measuring real-world RIRs offers several advantages: 1) microphone array speech can be generated by convolving RIRs with source signals, which exhibits sufficient data realness; 2) Direct-path speech can be easily obtained by convolving the direct-path impulse response (extracted from RIRs) with source signals, which can be used as the training target signal for speech enhancement; 3) information for source localization, such as the time-difference of arrival, can also be obtained from the multi-channel direct-path impulse responses. However, one significant drawback of RIR measurement is that it is more time-consuming than speech recording. Consequently, existing datasets such as MIR, BUTReverb, Reverb, ACE, and dEchorate offer only a limited range of measured RIRs and scenes. As for moving source, to obtain the RIR at densely-distributed discrete locations along a moving trajectory, the measurement process becomes even more time-consuming.

Other datasets directly provide multi-channel speech recordings. However, annotating the direct-path speech (as the training target signal) for speech enhancement poses challenges. Although some datasets provide close-talking signals, these cannot serve as the target signals since the direct-path speech is essentially an energy-attenuated and time-shifted version of the close-talking signals. To obtain a clean target signal, the CHiME-3 dataset [22] simulates the time delay of the direct-path speech for training, and also provides the speech signals recorded in a booth for development and test. In addition to evaluating the speech quality, speech enhancement can also be evaluated in terms of ASR performance. LibriCSS [23], MC-WSJ-AV [24], CHiME-5/-6/-7 [25], AMIMeeting [26], AISHELL-4 [27] and AliMeeting [28] (see Appendix for details) provide speech transcriptions for evaluating the speech enhancement performance via ASR. Due to the lack of target signal, the speech enhancement network for these datasets can only be trained with simulated data.

The annotation of sound source location is also not easy. VoiceHome-2 [29], LOCATA [30] and STARSS22/23 [31, 32] record speech of real human speakers and provide the speaker location. In VoiceHome-2, speakers can only locate at a small number of pre-defined positions, and their coordinates were measured with a laser telemeter. In LOCATA and STARSS22/23, speaker locations are obtained through an optical tracking system. However, the difficulty of setting up the optical tracking system limits the number of recording scenes.

**Multi-channel noise recording.** Noise recordings are typically provided either separately [33], or together with RIRs [21, 17, 18, 20, 19] or speech recordings [22]. However, most of these datasets [21, 17, 18, 20, 22, 19, 33] are limited in both quantity and diversity. Although the DEMAND dataset [33] offers noise signals recorded in various scenes with a 16-channel microphone array, but the duration of its recording is quite short.

By comparison, **the proposed RealMAN dataset** has the following advantages. *1) Realness.* Speech and noise are recorded in real environments. Direct recording for moving sources avoids issues associated with the piece-wise generation method. Different individuals move the loudspeaker freely to closely mimic human movements in real applications. *2) Quantity and diversity.* We record both speech signals and noise signals across various scenes. Compared with existing datasets, our collection offers greater diversity in spatial acoustics (in terms of acoustic scenes, source positions and states, etc) and noise types. This enables effective training of speech enhancement and source localization networks. *3) Annotation.* We provide detailed annotations for direct-path speech, speech transcriptions and source location, which are essential for accurate training and evaluation. *4) Number of channels.* The number of microphone channels, i.e. 32, is higher than almost all existing datasets, which facilitates the training of variable-array networks. *5) Relatively low recording cost.* The recording, playback, and camera devices are portable and easily transportable to different scenes.

Table 1: Existing microphone array speech datasets with speech enhancement and/or source localization annotations.

| Dataset | Diversity, Quantity | | | | Main Data | Microphone Array (×1 by default) |
|---|---|---|---|---|---|---|
| | # scene | scene type | source state | # RIR / speech duration | | |
| MIR [16] | 3 | Lab | Static | 78 RIRs | RIR | 8-ch linear (×3) |
| BUTReverb [17] | 9 | - | Static | 51 RIRs | RIR | 8-ch spherical |
| Reverb [18] | 3 | - | Static | 24 RIRs | RIR | 8-ch circular |
| DCASE [19] | 9 | Campus | Static | 38530 RIRs | RIR, location | 32-ch spherical |
| ACE [20] | 7 | Campus | Static | 14 RIRs | RIR, roomInfo | 2-ch, 3-ch triangle, 8-ch linear, 5-ch cruciform, 32-ch spherical |
| dEchorate [21] | 11 | Lab | Static | 99 RIRs | RIR, roomInfo | 5-ch linear (×6) |
| CHiME-3/-4 [22] | 5 | Multiple | - | 9.9 hours | Recording, transcription | 6-ch rectangular |
| VoiceHome-2 [29] | 12 | Home | Static | 5 hours | Recording, location, transcription | 8-ch |
| LOCATA [30] | 1 | Lab | Static, moving | 0.9 hours | Recording, location | 15-ch planar, 32-ch spherical, 12-ch robot, 4-ch hearing aids |
| STARSS22 [31] | 16 | Campus, corporation | Moving | 6.9 hours | Recording, location, sound event labels | 32-ch spherical |
| STARSS23 [32] | 16+ | Campus, corporation | Moving | 10.9 hours | Recording, location, sound event labels | 32-ch spherical |
| RealMAN (prop.) | 32 | Multiple | Static+ moving | 83.7 hours | Recording, location, direct-path signal, transcription | 32-ch (include various sub-arrays) |

# RIR involves room conditions, source positions and array positions
RoomInfo. denotes room acoustic information like reverberation time T60, direct-to-reverberant ratio (DRR), etc.
Only compact arrays are considered, and single microphone and close-talking (lapel and headset microphones) are excluded

Table 2: Existing microphone array noise datasets.

| Dataset | Noise Scene / Type | Duration |
|---|---|---|
| BUTReverb [17] | 9 rooms (large, middle and small size), with room environmental noise (silence) | 4.7 hours |
| Reverb [18] | 3 rooms (large, medium, and small), with stationary background noise mainly caused by air conditioning | 0.8 hours |
| ACE [20] | 7 offices, with meeting and teaching rooms, babble, ambient and fan noise recorded in each room | 13.6 hours |
| dEchorate [21] | 1 room with 11 surface absorptions, with diffuse babble, white and silence noises | 0.6 hours |
| CHiME-3/-4 [22] | 4 public scenarios (cafe, street junction, public transport and pedestrian area) | 8.4 hours |
| DCASE [19] | 9 rooms in university buildings, with ambient noise | 3.9 hours |
| DEMAND [33] | 18 scenes (domestic, office, public, transportation, nature categories) | 1.5 hours |
| RealMAN (prop.) | 31 scenes (indoor, semi-outdoor, outdoor, transportation categories) | 144.5 h |

DEMAND uses a 16-ch array in 4 staggered rows. The microphone array configurations for these datasets are listed in Table 1.

## 3 RealMAN Dataset

### 3.1 Recording system

Fig. 1a shows the recording system used in this work, which mainly consists of a 32-channel microphone array, a high-fidelity monophonic loudspeaker and a 360-degree fisheye camera.

**32-channel microphone array** is comprised of 32 high-fidelity omni-directional Audio-Technica BP899 microphones. The microphone's frequency response range is 20 Hz to 20 kHz. The array geometry is shown in Fig. 1b. This array encompasses the array topology found in common use cases, including common planar linear arrays, circular arrays, and 3D arrays. The sampling rate of microphone recording is 48 kHz. The sampled audio signals are then digitized by 4 clock-synchronized 8-channel microphone pre-amplifiers (RME OctoMic II) and processed by a laptop through an audio interface (RME Digiface USB).

**360-degree fisheye camera**: A 360-degree fisheye camera (HIKVISION DS-2CD63C5F-IHV) is placed right above the microphone array. The camera records the 360-degree panoramic image in real time synchronized with the microphone recording. The frame rate of the fisheye camera is 100 ms.

**High-fidelity monophonic speaker**: A high-fidelity monophonic loudspeaker (FOSTEX 6301 NE) is used to play source speech signals. The loudspeaker's frequency response range is 70 Hz to 15 kHz. It is placed on a height-adjustable and mobile carrier such that one can control the position of the loudspeaker to mimic a standing/moving human speaker. A 5-cm diameter LED light is put on the top of the loudspeaker to magnify the visibility of loudspeaker to the the fisheye camera and

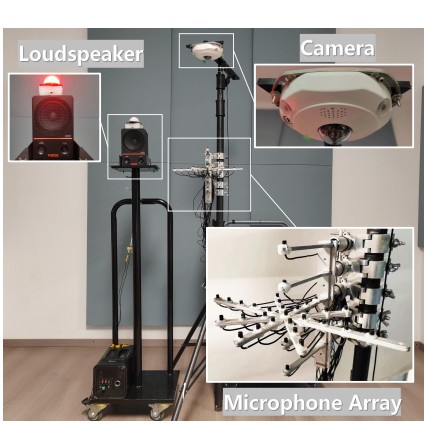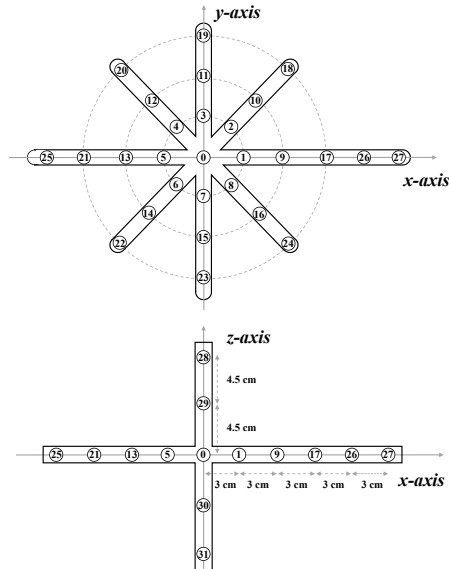

(a) Recording devices.       (b) The geometry of 32-channel microphone array.

Figure 1: Recording devices.

annotate the position of the loudspeaker. The LED light can emit red or green light, which is visible for the fisheye camera under various light conditions.

## 3.2 Source speech signals

Source speech signals that are played by the loudspeaker contains nearly 35 hours of clean Mandarin speech (close-sourced), of which about 30 hours are free-talking and 5 hours are reading. For free talk, speakers are encouraged to converse alone. Reading speech entail speakers reading news articles. The topics of speech content spread a wide range of domains including news reports, games, reading experiences, and life trivia. There are 55 speakers in total, including 27 males and 28 females. 17 speakers are recorded in a studio (T60 is smaller than 150 ms) with a high-fidelity microphone, while the rest are recorded in a living room with a bit larger reverberation time (T60 is about 200 ms) using a lower-fidelity microphone. The distance between speaker and microphone is about 0.2 m.

## 3.3 Speech and noise recording process

**Speech.** The objective of the speech recording process is to mirror the human activities in real speech applications, such as (multi-party) human-robot interaction, smart loudspeaker, meeting, etc. In each scene, the position of both the camera and microphone array are fixed. When playing source speech, the position of the loudspeaker takes on either static or moving states. For the moving case, one person manually moves the loudspeaker carrier with varying but reasonable moving speed. In transportation scenarios, people typically maintain a stationary position, thereby the loudspeaker only takes the static state. The height of the microphone array is always set to 1.40 m. In each scene, the microphone array is located at one position around the scene center part, and sometimes it is put close (not very close) to one side wall for having a large source-to-array distance or not disturbing other people. The center height of the loudspeaker is aligned with the height of the mouth of a standing person, varying randomly between 1.30 m and 1.60 m. Speech data was recorded across 32 distinct environments, including indoor, outdoor, semi-outdoor, and transportation scenarios. We ensure that most speech recordings were conducted under quiet conditions (usually at midnight). Over all scenes, the sound pressure level (SPL) of speech recordings and silent backgrounds are averagely about 68 dB (61 dBA) and 57 dB (36 dBA), respectively.

In different speech applications, speakers could have very different states in terms of facing or not facing the device, static or moving, small-range pacing or large-range walking, and head turning. For example, as for human-robot interaction and smart loudspeaker, the speaker normally faces the

device and could have some movements. In a meeting application, the speakers normally face each other and have head turning. During speech recording, we do consider these factors to a certain extent. For static speaker, most of the time, the loudspeaker faces towards the microphone array, with a small portion of side-facing (but no back-facing). For moving speaker, movements include a large portion of large-range walking, a small portion of small-range pacing and a smaller portion of head turning. The moving trajectory of large-range walking can be axial, tangential or in between, and can be random walking as well. During movement, the loudspeaker orientation is put either facing (most of the time side-facing) the microphone array, or towards the moving direction. The source-to-array distances are mainly distributed in the range of 0.5 m $\sim$ 5 m. Overall, in the proposed dataset, a number of speaker states are considered. However, we think it is still far from exhausting the speaker states in various speech applications, and the influence of speaker states on speech enhancement and source localization would be an interesting topic for future research.

The detailed speech recording information for each scene is given in Appendix B.1, including the recording duration, SPLs, room size and reverberation time. The statistics of source locations and several examples of speaker moving trajectory are given in Appendix B.4.

**Noise.** Noise recording is simpler, for which we place the microphone array in various environments to capture the real-world ambient noise. Noise recording is normally conducted in the daytime with active events in each environment. The collected recording clips with noise power lower than a certain threshold are abandoned. Then, an advanced voice activity detection is conducted to further filter out those recording clips including prominent speech signals. Noise is recorded in 31 different scenarios and ultimately retained 144.5 hours of recordings, covering most everyday scenarios. The SPL of noise recordings is averagely about 71 dB (58 dBA) over all scenes. The duration and SPLs of noise recording for each scene are given in Appendix B.1 as well.

### 3.4 Data annotation

**Direct-path target clean speech**. Deep learning-based speech enhancement methods require a target clean signal for training. Normally, the direct-path speech signal is used [34, 35]. For real-recorded datasets, before, providing direct-path speech has never been solved in the field.

In this paper, we develop a method to estimate the direct-path speech based on the source speech (replayed speech) and microphone recordings. The recording process is formulated in the time domain as $x(t) = s_{dp}(t) + s_{rev}(t) + n(t)$, $s_{dp}(t) = h_{dp}(t) * h_{dev}(t) * s(t) = h_{dev} * [As(t - \tau)]$, where $x(t), s_{dp}(t), s_{rev}(t)$ and $n(t)$ are the microphone recording, direct-path speech, speech reverberation, and noise, respectively. Theoretically, the direct-path speech is the convolution of the played source speech $s(t)$, the impulse response of the playing and recording devices $h_{dev}(t)$, and the direct-path impulse response $h_{dp}(t)$, where the direct-path impulse response $h_{dp}(t)$ can be formulated as a level attenuation $A$ and a time shift $\tau$ of $s(t)$. Note that $A$ and $\tau$ are time-invariant for static source, while time-varying for moving source.

The impulse response of devices (microphone and loudspeaker combined) $h_{dev}$ is considered to be constant (independent to the direction and orientation of the loudspeaker relative to the microphone) and it is measured in advance for one configuration where the loudspeaker faces toward the microphone with a source-to-microphone distance of 1 m, please refer to Appendix B.3 for more details. This simplified measurement is accurate for the omni-directional microphone (except for a constant time delay and level factor), but inaccurate for the frontal-directional loudspeaker as its impulse response is orientation-dependent. However, in our setting, it is difficult to annotate the loudspeaker orientation and thus to take the loudspeaker directivity into account, which is left for future research.

Then, the estimation of $s_{dp}(t)$ amounts to the estimation of $A$ and $\tau$ according to the known $x(t)$, $h_{dev}$, and $s(t)$. Note that, any constant time delay and level factor, such as the one caused by the measurement of $h_{dev}$ or by the loudspeaker orientation, will be absorbed into the estimated $\tau$ and $A$, respectively. For details of the estimation algorithm, please refer to Appendix C.1. The direct-path target speech can be estimated in the same way for all microphone channels. To keep a small data size, only the direct-path target speech for microphone 0 is included in the released dataset. In the future, we can provide more if there is a high requirement.

Due to the lack of ground truth, it is not straightforward to evaluate the estimation accuracy. We think the credibility of the estimated direct-path speech can be well testified based on the following criteria. i) The estimated direct-path speech should have the same speech quality as source speech

Table 3: Statistics of the training, validation, and test sets of RealMAN.

|  | Training | Validation | Test |
|---|---|---|---|
| Speech duration (hour) | 64.0 | 8.1 | 11.6 |
| - Moving speaker (hour) | 27.1 | 3.5 | 4.8 |
| - Static speaker (hour) | 36.9 | 4.6 | 6.8 |
| Noise duration (hour) | 106.3 | 16.0 | 22.2 |
| Number of scenes | 40 | 17 | 21 |
| Number of female speakers | 22 | 4 | 2 |
| Number of male speakers | 21 | 2 | 4 |

to provide an ideal upper-bound for speech enhancement. We have conducted an informal listening test by eight listeners to test whether there is an audible difference between the perceptual quality of the estimated speech and source speech, which shows that there is no audible difference with a 91% confidence. We have also tested the ASR performance, with an established ASR model trained on over 10,000 hours of Mandarin dataset WenetSpeech [36], where the results are identical for the estimated direct-path speech and source speech. ii) As shown in Appendix C.1, the estimation of $A$ and $\tau$ is based on the cross-correlation method. The good properties of intermediate results, such as the sharp correlation peak and the smooth estimation curve for moving source, also give us strong confidence on the estimation accuracy. iii) After all, as long as speech enhancement networks can be successfully trained with the estimated direct-path speech as the target (will be shown in the experiment section), the estimated direct-path speech can be deemed to be sufficiently accurate.

**Sound source location**. The annotation of source/loudspeaker location leverages a fisheye camera and an LED light (placed on top of the loudspeaker). During the speech recording process, the fisheye camera is placed right above the microphone array, and the plane coordinates of the camera and microphone array are aligned. The height of loudspeaker center (varying from 1.3 m to 1.6 m) is manually logged in the recording process. Given the pixel position of the LED light in the camera image, the source location can be calculated in three steps. i) Based on camera calibration, the azimuth angle (relative to both the array and camera) and elevation angle (relative to only the camera) of source can be calculated with the pixel position. ii) With the height and elevation angle (relative to the camera) of source, the horizontal distance between the source and camera/array can be calculated. iii) The elevation angle (relative to the array) of source can be calculated using the height of microphone array, the height and the horizontal distance of source. Note that, the fixed 12 cm height difference between the LED light and loudspeaker center has been taken into account.

To LED light is detected for each video frame with a vision-based detection algorithm. To guarantee the accuracy, all the detection results are manually double-checked. Please refer to Appendix C.3 for examples and pseudocodes of LED detection. According to the frame rate of the fisheye camera, the temporal resolution of source location annotation is set to 100 ms.

## 3.5   Dataset split and statistics

The recorded speech and noise data are split into training, validation, and test sets for learning-based speech enhancement and source localization, according the acoustic characteristics of the recording scenes and speaker identities:

**Acoustic characteristics of scenes**. Different scenes have different RIRs and noise characteristics. To make sure that the model can be trained under diverse scenes, there are 40 different scenes (of speech and noise) included in the training set. Various types of acoustic scenes are also provided in the validation and test sets (17 and 21, respectively), such that the algorithms can be fully evaluated under various scenarios. There are 3 scenes that only appeared in the test and validation sets, respectively, to further evaluate the generalization capability on the unseen acoustic scenes. We think the scene diversity of training and test sets are both critical for the full evaluation of general-purpose speech enhancement and source localization methods, so we have to make some scene overlaps across sets, but note that there is no data sample overlap among them.

**Speaker identities**. Following the general speech corpus split, the entire 55 speakers are split into 43, 6, and 6 for the training, validation, and test sets, respectively, without speaker overlap across sets. Please refer to Appendix B.2 for detailed speech duration statistics across speakers.

**Speech and noise matching**. During recording, speech and noise are normally recorded in quiet midnight and noisy daytime, respectively. To make the noisy speech as real as possible, it is better to mix speech and noise from the same scene. This principle is followed for the validation and test sets.

The number of scenes with both speech and noise recordings is 10 out of 17 for validation scenes and 11 out of 21 for test scenes. The same type of indoor environments, such as living rooms or office rooms, have similar noise characteristics, hence noise is not recorded for every scene. For these cases, we mix the speech of one scene with noise from a similar environment. Specifically, speech of all ClassRooms are mixed with ClassRoom1 noise; speech of all OfficeRooms and Library are mixed with OfficeRoom1/3 noise; speech of all LivingRooms are mixed with LivingRoom1 and Laundry noise. As for training, some preliminary experiments show that such match on speech and noise scenes is not required. Instead, a random scene match between speech and noise is more suitable for network training, possibly because of the promotion of data diversity.

The statistics of the dataset are shown in Table 3. The total 83.7 hours of the recorded speech are divided into 64.0, 8.1, and 11.6 hours for training, validation, and test, respectively. The total 144.5 hours of noise data are divided into 106.3, 16.0 and 22.2 hours for training, validation and test, respectively.

## 4    Baseline Experiments

In this section, we benchmark the proposed dataset for speech enhancement and source localization. As presented in Section 3.5, the validation and test sets are generated by mixing speeches and noises from matched scenes, and the signal level of mixed speech and noise are kept unchanged as recorded to maintain their natural loudness. The speech-noise mixed validation and test sets have an average SNR about 0.0 dB and -0.8 dB, respectively. Please refer to Appendix B.5 for the SNR statistics of validation and test sets. The training set is generated by randomly mixing speeches and noises, with a speech-recording-to-noise-recording ratio (SNR) uniformly distributed in [-10, 15] dB. In Appendix D.3, we also provide the experimental results on the recorded speech without adding noise.

In this paper, we only perform single-speaker speech enhancement (denoising and dereverberation) and source localization. Nevertheless, we think the proposed dataset can also be used for the tasks of multi-speaker separation and localization. Multi-speaker signals can be generated by simply mixing the single-speaker signals recorded in the same scene, which will be highly consistent to the simultaneous recording of multiple speakers. One unreal factor is that the background noise in speech recordings will also be mixed, which we think is not problematic, as the SPL of background noise is low compared to the SPL of speech, i.e. 57 dB (36 dBA) versus 68 dB (61 dBA).

### 4.1    Baseline methods and evaluation metrics

**Speech enhancement.** One popular time-domain network, i.e. FaSNet-TAC [13], and one recently-proposed frequency-domain network, i.e. SpatialNet [34], are used for benchmarking the speech enhancement performance. The negative of scale-invariant signal-to-distortion ratio (SI-SDR) [37] is used as the loss function for training the two networks. For FaSNet-TAC, the best configuration reported in its original paper is used. For SpatialNet, to reduce the computational complexity, a tiny version is used, where the hidden size of the SpatialNet-small version reported in the paper [34] is further reduced from 96 to 48. SI-SDR (in dB), WB-PESQ [38], and MOS-SIG, MOS-BAK, MOS-OVR from DNSMOS [39] are used for measuring the performance of speech enhancement. The ASR performance are evaluated by the WenetSpeech [36] ASR model trained by over 10,000 hours of Mandarin dataset, implemented in the ESPNet toolkit. Character error rate (CER, in percentage) is taken as the ASR metric.

**Sound source localization.** Azimuth angle localization is performed. We adopt a convolutional recurrent neural network (CRNN) as one baseline system for sound source localization. The baseline CRNN comprises a 10-layer CNN and a 1-layer gated recurrent unit. The kernel size of convolutional layers are all $3 \times 3$, each convolutional layer is followed by an instance normalization and a rectified linear unit activation function. Max pooling is applied to compress the frequency and time dimensions after every two convolutional layers. This baseline CRNN is very similar to the CRNN network used in many sound source localization methods [40, 41, 42]. The spatial spectrum, with candidate locations of every $1°$ azimuth angle, is used as the learning target [10, 43, 44, 45]. A linear classifier with sigmoid activation is used to predict the spatial spectrum. A recently proposed sound source localization method, i.e. IPDnet [15], is also used a baseline system. The hidden size of the original IPDnet is reduced from 256 to 128. Candidate locations are also set as every $1°$ azimuth angle. IPD templates are computed with the theoretical time delays from candidate locations to microphones,

Table 4: Benchmark experiments of speech enhancement.

| Baseline | Training Data | | Static Speaker | | | | | | Moving Speaker | | | | | |
|---|---|---|---|---|---|---|---|---|---|---|---|---|---|---|
| | speech | noise | WB-PESQ | SI-SDR | MOS-SIG | MOS-BAK | MOS-OVR | CER | WB-PESQ | SI-SDR | MOS-SIG | MOS-BAK | MOS-OVR | CER |
| unprocessed | - | | 1.14 | -9.9 | 2.01 | 1.73 | 1.52 | 20.1 | 1.10 | -9.0 | 1.80 | 1.54 | 1.37 | 23.7 |
| FaSNet-TAC [13] | sim | sim | 1.39 | -2.4 | 2.71 | 3.00 | 2.18 | 25.4 | 1.34 | -2.5 | 2.62 | 2.92 | 2.10 | 28.2 |
| | sim | real | **1.47** | -1.0 | **2.83** | 3.24 | **2.36** | **20.5** | **1.41** | -0.8 | **2.75** | 3.12 | **2.27** | **23.4** |
| | real | sim | 1.42 | 0.4 | 2.63 | 3.21 | 2.19 | 24.0 | 1.36 | 0.4 | 2.51 | 3.15 | 2.08 | 28.2 |
| | real | real | 1.46 | **2.1** | 2.79 | **3.30** | 2.34 | 21.9 | 1.39 | **1.2** | 2.73 | **3.20** | 2.26 | 25.6 |
| SpatialNet [34] | sim | sim | 1.37 | -5.2 | **3.24** | 2.85 | 2.46 | 17.9 | 1.37 | -4.8 | **3.20** | 2.75 | 2.40 | 21.0 |
| | sim | real | 1.13 | -9.7 | 1.59 | 1.67 | 1.32 | 44.2 | 1.11 | -9.2 | 1.51 | 1.57 | 1.26 | 49.0 |
| | real | sim | 1.96 | 4.7 | 3.16 | 3.13 | 2.52 | 17.3 | 1.80 | 2.7 | 3.08 | 3.05 | 2.42 | 21.2 |
| | real | real | **2.16** | **7.3** | 3.23 | **3.43** | **2.71** | **14.5** | **1.97** | **4.3** | 3.15 | **3.40** | **2.63** | **18.6** |

following [15]. The training target of ground truth direct-path IPDs are computed with the annotated azimuth angles. The localization results are evaluated with i) the Mean Absolute Error (MAE) and ii) Localization Accuracy (ACC) ($N^\circ$), namely the ratio of frames with the estimation error of azimuth less than $N^\circ$. Please refer to Appendix D.1 for more detailed experimental configurations.

## 4.2 Benchmark experiments

Benchmark experiments use a 9-channel sub-array (the smallest circle and the center microphone, and the center microphone is used as the reference microphone when necessary). In addition, we also evaluate the effect of the simulation-to-real mismatch on speech enhancement and source localization tasks. Equal amounts of multichannel speech are simulated according to our real-recorded dataset, using the gpuRIR toolkit [46]. Specifically, one counterpart utterance is simulated for each real-recorded utterance using the same source speech, room size, T60, and source position/trajectory as the real-recorded utterance. The directivity of microphone and source are always set to be omni-directional during simulation. Multi-channel noise is simulated with the diffuse noise generator [5], taking white (code generated), babble and factory noise (from noisex-92 [47]) as the single-channel source noise. The simulated speech/noise can also be combined with real-recorded noise/speech. For each setting, the SNR for mixing speech and noise is uniformly sampled in [-10, 15] dB.

**Speech Enhancement.** The results of speech enhancement are shown in Table 4. Overall, compared to other settings, training with real speech and real noise achieves the best speech enhancement performance on the real-recorded test set, for both the baseline networks. As for intrusive metrics, i.e. WB-PESQ and SI-SDR, the target clean speech provided in this dataset are used as the reference signals, which may leads to some measurement bias for other settings, therefore these metrics are only presented for reference. The non-intrusive DNS-MOS scores can better reflect the speech quality. It can be seen that, for FaSNet-TAC, training with simulated speech and real noise achieves comparable DNS-MOS performance as the setting of real speech plus real noise, which indicates that speech simulation does not have the simulation-to-real problem for FaSNet-TAC. However, this is not the case for SpatialNet, for which training with simulated speech and real noise achieves the worst performance. Some validation experiments had been conducted to figure out the reasons for this phenomenon, which showed that there might be a slight mismatch between the real and ideal microphone positions. We have designed a new method for resolving the problem of microphone position mismatch, namely disturbing the ideal microphone positions in training, which is shown to be very effective on simulated test data, but only slightly improve the performance on our real test data. This indicates that there are still some unclear mismatches between the simulated and real data. As its name indicates, SpatialNet [34] mainly learns the RIR-related spatial information, which is possibly more sensitive to those unclear mismatches. Exploring and resolving those mismatches would be an interesting topic for future research.

Training with real speech and real noise consistently outperforms training with simulated noise. The simulated noise lacks diversity in terms of both spectral pattern and spatial correlation. The spatial correlation of real ambient noise could largely deviates from the one of theoretical diffuse noise field. Moreover, the spatial correlation of real noise is also highly time-varying. Please see Appendix D.2 for the detailed analysis of spatial correlation of real noise. The simulation-to-real mismatch of noise leads to the performance degradation. In addition, due to the high complexity and non-stationarity of the spatial correlation of real noise, it is complicated to develop new techniques for simulating real noise. Therefore, we suggest to use real-recorded multi-channel noise for training speech enhancement networks.

Table 5: CRNN benchmark experiments of sound source localization.

| Training Data | | Static Speaker | | Moving Speaker | |
|---|---|---|---|---|---|
| speech | noise | ACC(5°) [%] | MAE [°] | ACC(5°) [%] | MAE [°] |
| sim | sim | 60.4 | 7.8 | 58.8 | 9.6 |
| sim | real | 52.4 | 22.9 | 48.4 | 21.4 |
| real | sim | 80.0 | 5.7 | 75.8 | 8.4 |
| real | real | **89.2** | **2.2** | **86.7** | **3.1** |

Table 6: IPDnet variable-array experiments for sound source localization.

| Network | Static Speaker | | Moving Speaker | |
|---|---|---|---|---|
| | ACC(5°) [%] | MAE [°] | ACC(5°) [%] | MAE [°] |
| Fixed-Array | 87.0 | 3.0 | **85.4** | **3.1** |
| Variable-Array | **87.1** | **2.4** | 78.8 | 3.7 |

Table 7: FaSNet-TAC variable-array experiments for speech enhancement.

| Network | Static Speaker | | | | | | Moving Speaker | | | | | |
|---|---|---|---|---|---|---|---|---|---|---|---|---|
| | WB-PESQ | SI-SDR | MOS-SIG | MOS-BAK | MOS-OVR | CER | WB-PESQ | SI-SDR | MOS-SIG | MOS-BAK | MOS-OVR | CER |
| unprocessed | 1.14 | -9.9 | 2.01 | 1.73 | 1.52 | 20.1 | 1.10 | -9.0 | 1.80 | 1.54 | 1.37 | 23.7 |
| Fixed-Array | 1.41 | **1.5** | **2.76** | 3.25 | **2.28** | **25.5** | **1.36** | **0.5** | **2.72** | 3.18 | **2.23** | **30.0** |
| Variable-Array | **1.42** | 1.0 | 2.70 | **3.33** | **2.28** | 27.4 | **1.36** | 0.3 | 2.64 | **3.27** | 2.20 | 31.4 |

Overall, the proposed dataset is a difficult one for speech enhancement, due to the large scene diversity, the high realness, and the complex acoustic conditions. The CERs of unprocessed recordings are quite high, i.e. close to or larger than 20%, even though a very strong ASR model (trained with over 10,000 hours data) is used.

**Sound source localization.** The results of the sound source localization are presented in Table 5. The mismatch between simulated RIRs and real recordings causes the performance degradation of sound source localization, which is consistent with the findings in [12, 48, 49]. The simulation-to-real mismatch of noise causes the performance degradation of sound source localization as well.

### 4.3 Variable-array networks and array generalization

End-to-end speech enhancement and source localization networks are normally array-dependent, which means although the fixed-array networks (presented in the previous section) trained using real speech and real noise achieve better performance for one array, they still cannot be used for other arrays. In this section, we use all the 28-microphone data (microphone 0 ∼ 27) on the horizontal plane to train the variable-array networks, i.e. FaSNet-TAC [13] for speech enhancement and IPDnet [15] for source localization, to see whether the trained networks can be directly used to unseen arrays.

We set one test array, namely a 5-channel uniformly-spaced linear array (microphone 11, 3, 0, 7, and 12). The training of variable-array networks uses randomly selected 2 ∼ 8-channel sub-arrays, excluding all 5-channel uniformly-spaced linear arrays. The microphone 0 is always used and taken as the reference channel.

Table 7 and Table 6 present the results of speech enhancement and sound source localization, respectively. It can be seen that there are indeed certain performance losses when compared with the fixed-array networks that are trained using the test array, but the losses are relatively small. This shows that the 32-channel real-recorded microphone array data provided in the proposed dataset can successfully train the variable-array networks, which offers a competitive solution for real-world multi-channel speech enhancement and sound source localization.

## 5 Conclusion

This paper presents a new real-recorded and annotated microphone array speech and noise dataset, named **RealMAN**, for speech enhancement and source localization. Baseline experiments demonstrate that training with our real-recorded data outperforms training with simulated data, by eliminating the simulation-to-real gap. The performance on our dataset can better reflect the capabilities of tested algorithms in real-world applications, providing a more reliable benchmark for speech enhancement and source localization. Additionally, variable-array networks can be successfully trained using various sub-arrays of the proposed 32-channel microphone array, and they have the potential to be applied directly to unseen arrays in real-world applications.

## Acknowledgments and Disclosure of Funding

This work was supported by the Zhejiang Provincial Natural Science Foundation of China under Grant 2022XHSJJ008.

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

# A  Existing Microphone Array Speech Datasets

As a supplement to Table 1, additional existing datasets of microphone array speech recordings are listed in Table 8, which includes LibriCSS [23], MC-WSJ-AV [24], CHiME-5/-6/-7 [25], AMIMeeting [26], AISHELL-4 [27] and AliMeeting [28]. Though these datasets provide multi-channel speech recordings with speech transcriptions, they don't provide any direct annotations for speech enhancement and source localization.

Table 8: Additional microphone array speech datasets.

| Dataset | Diversity, Quantity | | | | Main Data | Microphone Array (×1 by default) |
|---|---|---|---|---|---|---|
| | # scenes | scene type | source state | speech duration | | |
| LibriCSS [23] | 1 | Meeting | Static | 10 hours | Recording, transcription | 7-ch circular |
| MC-WSJ-AV [24] | 3 | Meeting | Static+moving | - | Recording, transcription | 8-ch circular (×2) |
| CHiME-5/-6/-7 [25] | 20 | Home | Moving | 50 hours | Recording, transcription | Kinect (×6), binaural pairs (×4) |
| AMIMeeting [26] | 3 | Meeting | - | 100 hours | Recording, transcription | 8-ch circular, 8/4/10-ch |
| AISHELL-4 [27] | 10 | Meeting | Static+ slightly moving | 120 hours | Recording, transcription | 8-ch circular |
| AliMeeting [28] | 21 | Meeting | Static | 120 hours | Recording, transcription | 8-ch circular |

# B  Dataset Details

## B.1  Scene information

We present detailed information of acoustic scenes and recordings for RealMAN in Table 9. The scene names in this table are consistent with the names in our released datasets. For each scene, the duration of static/moving speech and noise, the sound pressure level of speech/background (during speech recording) and noise, the reverberation time (T60) of indoor and semi-outdoor scenes, the room size when available are provided. The T60s of most enclosed scenes are measured using our recording system with the exponential sine sweep signal (see details in Appendix B.3). For some scenes that are not feasible for our T60 measurement, the T60s are measured by a mobile phone application, which however could be inaccurate.

## B.2  Speech duration statistics across speakers

The speech duration statistics of different speakers in the dataset are given in Fig. 2.

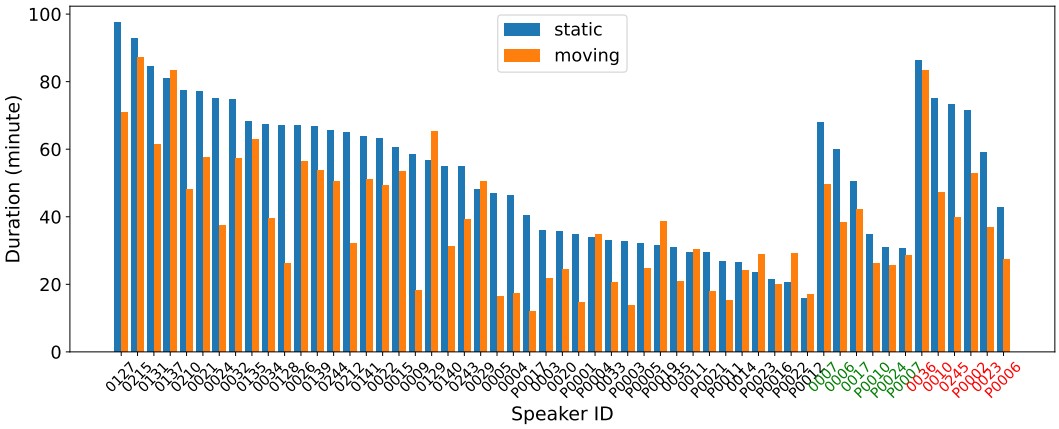

Figure 2: Speech duration statistics across speakers in the RealMAN dataset. In the naming of speaker ID, the one beginning with 'P' denotes speaker in read speech, and the other denotes speaker in free talk. The speaker ID is color-coded to distinguish the train, validation, and test sets.

Table 9: Summary of recording scene information. The T60s of indoor scenes are measured by ourselves (see Appendix B.3 for measurement details), except that the underlined ones are measured by a less precise phone application. The scene types are abbreviated as follows: semi. for semi-outdoor and trsp. for transportation.

| Scene | | Duration of speech and noise in train/validation/test sets (minute) | | | Sound pressure level (dB/dBA) | | | T60 (s) | Room Size (m) |
|---|---|---|---|---|---|---|---|---|---|
| name | type | static | moving | noise | speech | background | noise | | |
| BasketballCourt1 | Outdoor | 34 / 10 / 39 | 40 / 8 / 42 | 315 / 52 / 158 | 68 / 60 | 63 / 47 | 69 / 53 | - | - |
| Car (Electric) | Trsp. | 51 / 42 / 23 | - | 172 / 129 / 129 | 71 / 63 | 61 / 35 | 77 / 48 | 0.09 | - |
| OfficeRoom3 | Indoor | 68 / 3 / 5 | 81 / 7 / 8 | 214 / 72 / 72 | 70 / 63 | 56 / 29 | 63 / 44 | 1.30 | 8.5×6.0×3.5 |
| OfficeLobby | Indoor | 47 / 29 / 5 | 51 / 28 / 8 | 144 / 42 / 21 | 64 / 57 | 56 / 38 | 61 / 50 | 2.48 | - |
| SunkenPlaza1 | Semi. | 81 / 11 / 9 | 68 / 15 / 8 | - | 67 / 60 | 64 / 45 | - | 0.92 | 27.6×14.6 |
| BadmintonCourt1 | Indoor | 77 / 13 / - | 40 / 7 / - | 317 / 80 / - | 61 / 55 | 50 / 27 | 70 / 58 | 1.58 | 28.6×7.7×8.6 |
| Gym | Indoor | 73 / 1 / - | 71 / 6 / - | 293 / 33 / - | 64 / 59 | 45 / 28 | 61 / 53 | 1.50 | 39.5×8.5×4.4 |
| Market | Indoor | 90 / 9 / - | 128 / 8 / - | 255 / 29 / - | 67 / 60 | 60 / 39 | 71 / 67 | 1.23 | 31.0×30.0×4.9 |
| Auditorium | Indoor | 123 / 14 / - | 73 / 11 / - | - | 62 / 56 | 50 / 32 | - | 1.18 | |
| LivingRoom6 | Indoor | 54 / 15 / - | 51 / 18 / - | - | 69 / 64 | 47 / 28 | - | 0.40 | 3.6×2.0×2.6 |
| LivingRoom8 | Indoor | 68 / 22 / - | 49 / 14 / - | - | 73 / 68 | 54 / 33 | - | 0.40 | 3.8×2.0×2.6 |
| Cafeteria1 | Indoor | 76 / - / 40 | 45 / - / 22 | 483 / - / 120 | 64 / 57 | 55 / 31 | 66 / 58 | 0.76 | - |
| Car (Gasoline) | Trsp. | 100 / - / 17 | - | 189 / - / 81 | 69 / 64 | 58 / 34 | 92 / 63 | 0.07 | - |
| Classroom2 | Indoor | 101 / - / 14 | 81 / - / 7 | - | 66 / 57 | 62 / 33 | - | 0.68 | 20.0×16.4×3.5 |
| Classroom3 | Indoor | 86 / - / 39 | 55 / - / 26 | - | 68 / 61 | 55 / 43 | - | 1.36 | 23.8×6.9×3.9 |
| Library | Indoor | 100 / - / 26 | 69 / - / 21 | - | 66 / 59 | 57 / 44 | - | 1.18 | radius=8.8 |
| LivingRoom2 | Indoor | 8 / - / 9 | 9 / - / 7 | - | 70 / 66 | 44 / 29 | - | 0.84 | 4.5×2.8×2.9 |
| LivingRoom4 | Indoor | 200 / - / 36 | 147 / - / 26 | - | 70 / 63 | 56 / 36 | - | 0.43 | 7.7×3.0×2.0 |
| LivingRoom5 | Indoor | 90 / - / 10 | 84 / - / 10 | - | 71 / 63 | 52 / 33 | - | 0.56 | 4.2×3.6×2.9 |
| OfficeRoom4 | Indoor | 80 / - / 14 | 60 / - / 11 | - | 69 / 63 | 63 / 41 | - | 0.89 | 6.8×5.1×3.3 |
| Bus(Electric) | Trsp. | 109 / - / - | 7 / - / - | - | 71 / 65 | 64 / 43 | 78 / 63 | 0.28 | - |
| Classroom1 | Indoor | 76 / - / - | 72 / - / - | 338 / 25 / 120 | 69 / 60 | 64 / 31 | 65 / 40 | 0.82 | 12.7×7.6×3.5 |
| LivingRoom1 | Indoor | 81 / - / - | 69 / - / - | - / - / 7 | 68 / 62 | 48 / 32 | 68 / 60 | 0.60 | 9.7×9.6×3.0 |
| LivingRoom3 | Indoor | 88 / - / - | 99 / - / - | - | 69 / 62 | 54 / 34 | - | 0.38 | 8.5×5.3×2.6 |
| LivingRoom7 | Indoor | 132 / - / - | 84 / - / - | - | 67 / 61 | 59 / 38 | - | 0.40 | 12.9×8.2×2.2 |
| UndergroundParking1 | Indoor | 125 / - / - | 103 / - / - | - | 68 / 60 | 56 / 39 | - | 2.92 | - |
| BasketballCourt2 | Outdoor | - / 33 / - | - / 28 / - | - / 166 / - | 69 / 63 | 62 / 47 | - | 0.84 | - |
| Cafeteria3 | Indoor | - / 34 / - | - / 25 / - | - / 286 / - | 67 / 61 | 59 / 40 | 69 / 59 | 0.84 | - |
| OfficeRoom1 | Indoor | - / 39 / - | - / 36 / - | - | 70 / 64 | 65 / 34 | - | 0.72 | 18.6×10.5×5.0 |
| BadmintonCourt2 | Indoor | - / - / 43 | - / - / 28 | - / - / 355 | 62 / 57 | 52 / 29 | 67 / 62 | 1.69 | 28.6×7.7×8.6 |
| OfficeRoom2 | Indoor | - / - / 38 | - / - / 37 | - / - / 118 | 66 / 60 | 59 / 41 | 66 / 46 | 0.49 | - |
| UndergroundParking2 | Indoor | - / - / 43 | - / - / 27 | 233 / - / 100 | 67 / 62 | 59 / 44 | 67 / 51 | 4.93 | - |
| BarberShop | Indoor | - | - | 213 / - / - | - | - | 69 / 64 | 0.65 | - |
| Cafeteria2 | Indoor | - | - | 443 / - / - | - | - | 72 / 63 | 1.24 | 13.3×9.2×3.7 |
| ConstructionPlant | Outdoor | - | - | 346 / - / - | - | - | 69 / 58 | - | - |
| Laundry | Indoor | - | - | 28 / 7 / 12 | - | - | 64 / 55 | 0.53 | 3.6×2.6×2.5 |
| LivingRoom9 | Indoor | - | - | 45 / - / - | - | - | 64 / 59 | 0.56 | 9.7×9.6×3.0 |
| Park | Outdoor | - | - | 338 / - / - | - | - | 73 / 59 | - | - |
| PedestrianStreet | Semi. | - | - | 495 / - / | - | - | 73 / 61 | 0.34 | - |
| Roadside1 | Outdoor | - | - | 169 / - / - | - | - | 75 / 62 | - | - |
| Roadside2 | Outdoor | - | - | 163 / - / - | - | - | 76 / 63 | - | - |
| ShoppingMall | Indoor | - | - | 327 / - / - | - | - | 77 / 72 | 0.95 | - |
| Subway (Electric) | Trsp. | - | - | 177 / - / - | - | - | 86 / 71 | - | - |
| SunkenPlaza2 | Semi. | - | - | 167 / 36 / 36 | - | - | 66 / 48 | 0.92 | - |
| Terrace1 | Semi. | - | - | 239 / - / - | - | - | 73 / 55 | 0.36 | - |
| Terrace2 | Semi. | - | - | 26 / - / - | - | - | 72 / 59 | 0.61 | - |

## B.3 T60 and device impulse response measurement

T60s are measured based on the RIR measurement with an exponential sine sweep signal. Firstly, we generate repeated exponential sine sweeps with a frequency ranging from 200 Hz to 8 kHz and their inverse signals to measure the RIR [50]. Secondly, the energy decay curve of the measured RIR is calculated. The appropriate segment in the energy decay curve is selected to calculate the T20 using linear least-squares regression. The T60 is approximated as three times the T20. For each scene, we conduct multiple trials to measure the T60, and the mean value is taken as the final T60 measurement.

For measuring the device impulse response (i.e. $h_{dev}$), we also first measure the RIR. Then any one clearly distinct path (say the direct path) in the RIR encodes the device impulse response, which is extracted as the measurement of $h_{dev}$.

## B.4    Source location statistics

Fig. 3 shows the histogram of source azimuth, elevation and distance. The azimuth angles are mainly distributed in the range of $0 \sim 180°$. Sometimes, we need to put the microphone array close to one of the side walls for having a large source-to-array distance or not disturbing other people. To maintain the recording consistence, we decided to put the speaker mainly in the $0 \sim 180°$ space even when the array is not put close to one side wall. Due to the symmetry of the array, we think the $0 \sim 180°$ space is fairly sufficient for representing the whole space in terms of the evaluation of speech enhancement and source localization. The elevation angles are mainly distributed in the range of $-5 \sim 15°$ considering that normally the height of device and speaker are not distinct too much. The source-to-array distances are mainly distributed in the range of 0.5 m $\sim$ 5 m, covering the main range of most speech applications.

Fig.4 shows real examples of four typical types of speaker moving trajectory.

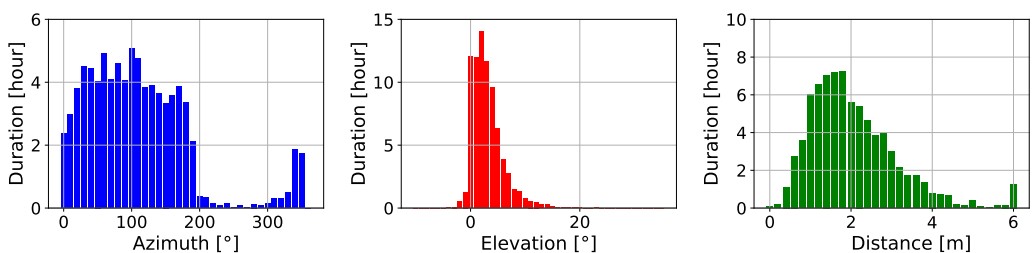

Figure 3: Histogram of speaker azimuth, elevation and distance.

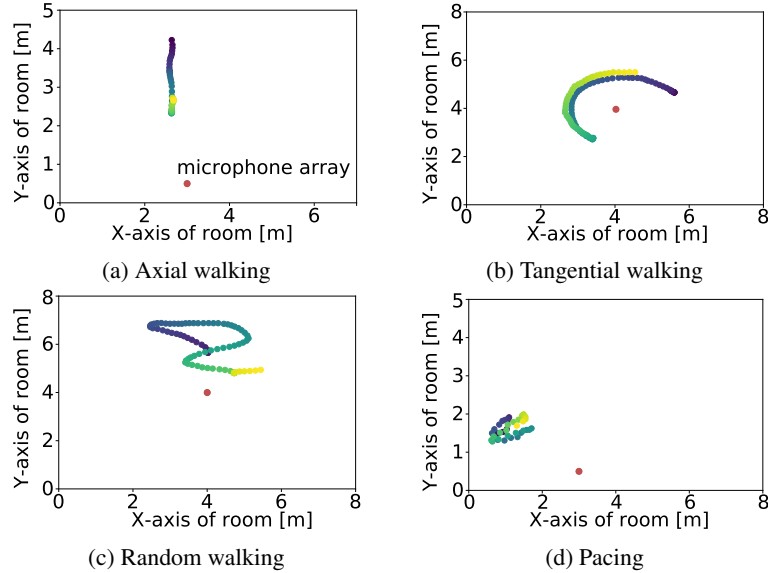

Figure 4: Four typical types of speaker moving trajectory. For each trajectory, the speaker height is constant, thus the moving trajectory is visualized only in the horizontal plane. Colors from lighter to darker indicate the time evolving.

## B.5    SNR distribution of validation and test sets

The SNR distribution of the speech-noise mixed validation and test sets are shown in Fig. 5, whose mean values are 0.0 dB and $-0.8$ dB, respectively.

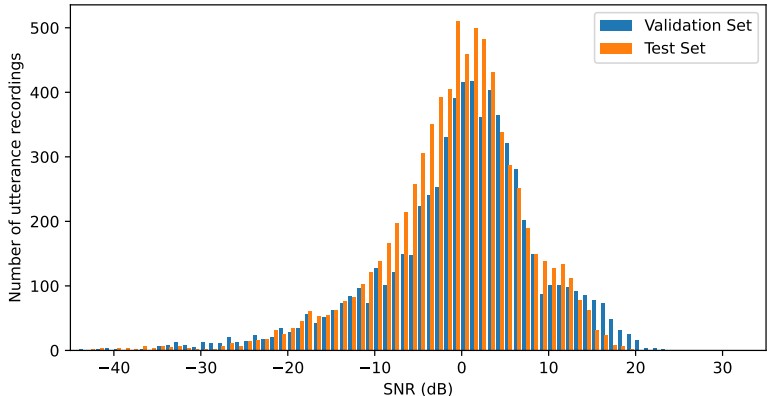

Figure 5: Histogram of SNR for validation and test sets.

## C   Annotation Details

### C.1   Details for target direct-path clean speech estimation

It is difficult to accurately estimate $A$ and $\tau$ from noisy and reverberant speech recordings. Therefore, we leverage a prior speech enhancement step (using the AlignSpatialNet presented in Appendix C.2) which gives a rough estimation of the direct-path speech, denoted as $\hat{s}_{dp}(t)$, which is supposed to be roughly time- and level-aligned with $s_{dp}(t)$. Note that $\hat{s}_{dp}(t)$ is not directly used as the wanted target clean speech, because its interpretability and accuracy are not good enough. Alternatively, we calculate the generalized cross-correlation (GCC) [51] between $h_{dev} * s(t)$ and $\hat{s}_{dp}(t)$ to estimate the time shift $\tau$ (at the sample level) by finding the maximum peak of GCC. For static sources, $A$ and $\tau$ are time-invariant, and we estimate them at the utterance level. As for moving sources, $A$ and $\tau$ are time-varying. Because GCC is a statistical value, the estimation at the sample level is intractable. Therefore, we clip one speech utterance into overlapping segments and estimate $\tau$ at the segment level. The segment length of direct-path speech is determined by the time delay estimate of consecutive segments, and then the segments of source speech are stretched or compressed (by the resampling operation) to align with the segments of direct-path speech. After that, the estimation of segment-level $A$ can also be done. Note that, data cleansing and nonlinear interpolation operations are adopted for both $\tau$ and $A$ estimations. After all, $\tau$ and $A$ estimates for moving sources are at the segment and sample levels, respectively. The details for target clean speech estimation of static and moving sources are shown in Algorithm 1 and 2, respectively. For simplicity, the sampling rate of the speech signal is marked in the parentheses and the downsampling operation is omitted.

Since there is no available ground truth, we illustrate an example of GCC in Fig. 6a, and an example of segment-level time shift and sample-level attenuation estimates in Fig. 6b for judging the credibility of the estimation of $A$ and $\tau$. The single sharp peak of GCC indicates the reliability of the time

---

**Algorithm 1:** Target clean speech estimation for static speaker

**Input:** Source clean speech utterance $s_{dp}(t)$, AlignSpatialNet output utterance $\hat{s}_{dp}(t)$.

1 Utilize the generalized cross-correlation (GCC) [51] between AlignSpatialNet output (8 kHz) and source clean speech (8 kHz) to estimate the time shift of the direct-path speech (8 kHz, 48 kHz);

2 Based on the power of AlignSpatialNet output (8 kHz) and time-aligned target clean speech (8 kHz) to estimate the attenuation of target clean speech;

3 Generate the target direct-path clean speech (48 kHz) with the estimated time shift and level attenuation;

4 Manually check the target clean speech (48 kHz) and corresponding recordings (48 kHz), and the speech utterance will be discarded if it is unreliable.

**Output:** Target direct-path clean speech utterance (48 kHz).

---

**Algorithm 2:** Target clean speech estimation for moving speaker

---

**Input:** Source clean speech utterance $s_{dp}(t)$, AlignSpatialNet output utterance $\hat{s}_{dp}(t)$.

1  Divide the speech utterance into 1-second segments with an overlap of 50%;
2  **for** *segment in the utterance* **do**
3      Utilize the GCC between segments in AlignSpatialNet output (8 kHz) and source speech (8 kHz) to estimate the time shift of the direct-path speech;
4  **end**
5  Cleanse the time shift estimates by removing unreasonable values;
6  Utilize the Pchip interpolator to replace the unreasonable time shift estimates and generate a smooth time shift sequence for all segments;
7  **for** *segment in the utterance* **do**
8      Based on the time shift estimates, stretch the source segment by resampling operation (to mimic the Doppler frequency shift effect caused by speaker movement) and generate the time-aligned direct-path speech segment (8 kHz, 48 kHz);
9      Based on the power of AlignSpatialNet output (8 kHz) and time-aligned direct-path speech segment (8 kHz) to estimate the attenuation of direct-path speech segment;
10  **end**
11  Cleanse the attenuation estimates by removing unreasonable values;
12  Utilize the Pchip interpolator to replace the unreasonable attenuation estimates and generate a smooth attenuation sequence for all sampling points;
13  **for** *sampling point in time-aligned direct-path speech (48 kHz)* **do**
14      Apply the attenuation estimates to the sampling point;
15  **end**
16  Concatenate the direct-path speech sampling points to generate the direct-path speech utterance (48 kHz);
17  Manually check the direct-path speech (48 kHz) and corresponding recordings (48 kHz), and the speech utterance will be discarded if it is unreliable.

**Output:** Target direct-path clean speech utterance (48 kHz)

---

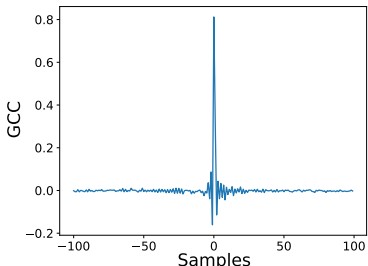
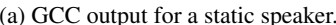
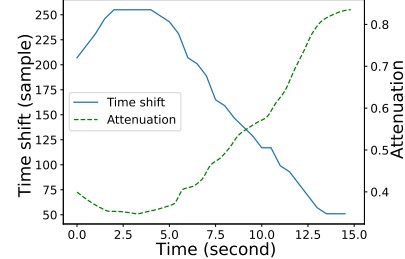

    (a) GCC output for a static speaker.     (b) Time shift and gain estimates for a moving speaker.

Figure 6: Examples of intermediate results for target direct-path clean speech estimation.

shift estimation. Note that if the GCC of one utterance for the static speaker (or one segment for the moving speaker) does not present such a sharp peak, the utterance for the static speaker will be abandoned (or the segment-level time shift estimation for the moving speaker will be abandoned, and recovered by interpolation). The smoothness and consistency of $A$ and $\tau$ estimates indicate the reliability of $A$ and $\tau$ estimates for moving speakers.

### C.2   Network architecture for source-signal guided direct-path speech estimation

We propose a network named AlignSpatialNet for source-signal guided direct-path speech estimation. As shown in Fig. 7, the network has two branches, one for the recorded microphone signals and one for the played source clean speech signal. Each branch is extended from our previously proposed SpatialNet [34]. Besides the neural components from SpatialNet, i.e. the cross-band block, the multi-head self-attention module (MHSA) and the time-convolutional feedforward network (T-ConvFFN), each branch also contains multi-head cross-attention module (MHCA) for extracting the spatial cues,

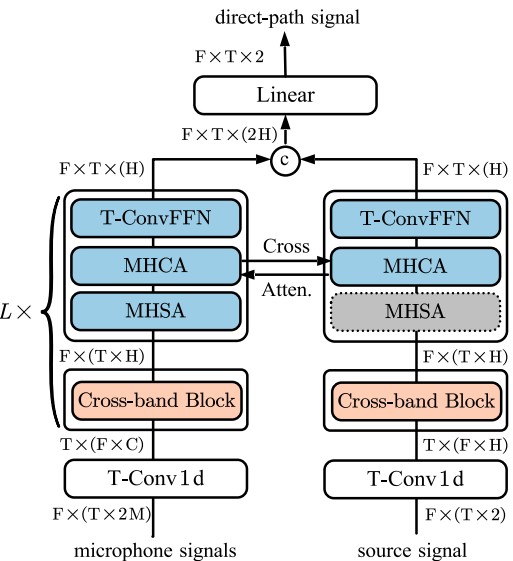

Figure 7: The architecture of the proposed AlignSpatialNet.

i.e. time shift and level attenuation, between the played source signal and recorded microphone signals.

The MHCA module takes the hidden representations of microphone signals denoted as $\mathbf{h}_m \in \mathbb{R}^{F \times T \times H}$ and source signal denoted as $\mathbf{h}_s \in \mathbb{R}^{F \times T \times H}$ as input, where $F$, $T$ and $H$ denote the number of frequencies, time frames, and hidden units. The cross-attention in MHCA can be formulated as:

$$\mathbf{h}_q[f, t, :] \leftarrow \text{Attention}(\mathbf{h}_q[f, t, :], \mathbf{h}_k[f, t - l : t - l, :], \mathbf{h}_v[f, t - l : t - l, :])$$

where $\mathbf{h}_q$, $\mathbf{h}_k$ and $\mathbf{h}_v$ are respectively the query, key and value vectors, which correspond to $\mathbf{h}_s$, $\mathbf{h}_m$ and $\mathbf{h}_m$ in the source branch and correspond to $\mathbf{h}_m$, $\mathbf{h}_s$ and $\mathbf{h}_s$ in the recording branch. $f$ and $t$ denote the frequency index and frame index, and $l$ is a preset value for the maximum number of frame shifts in one layer, which is set to a value corresponding to 200 ms in our experiment.

The proposed AlignSpatialNet is trained with simulated data. Note that, for moving sources, when the moving speed is high, the simulated direct-path signals may have some clicking noise and other audible artifacts [9]. To mitigate this problem, the speech signals at two adjacent locations are half-overlapped and applied with a trapezium window for both generating the reverberant signal and direct-path signal in moving case. The simulated reverberant microphone signals are added with our real-recorded multichannel noise signals with signal-to-noise ratio (SNR) sampled in $[5, 20]$ dB. In training, the negative of SNR is used as the loss function.

### C.3 Details for LED light detection with the fisheye camera

Algorithm 3 provides the pseudocode for the LED light detection algorithm, Fig. 8 shows four examples of the images captured by the fisheye camera and the LED light detection results (red boxes).

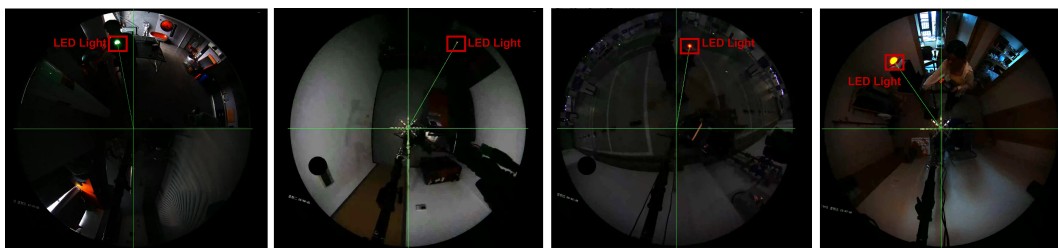

Figure 8: Examples of the vision-based LED light detection in different recording scenes.

**Algorithm 3:** Estimating Sound Source Position from Eyefish Camera Video

**Input:** Eyefish camera video

1 **for** *frames in video* **do**
2      Convert video frame to HSV domain;
3      Create a mask for red/green color regions;
4      Apply mask to the HSV frame;
5      Initialize a score map;
6      **for** *pixel in masked region* **do**
7          Within the score map, calculate the pixel score based on the color intensity and brightness;
8      **end**
9      Find the pixel of maximum score in the score map as the detected position of the LED light (sound source);
10 **end**

**Output:** Position of the sound source

# D    Experiments

## D.1    Details for experimental configuration

All speech enhancement and sound source localization networks are trained on 16 kHz 4-second speech clips, and tested on 16 kHz varying-long speech utterances.

For speech enhancement networks, the training configurations are set the same as in their original papers. Each model is trained for 50 epochs, and the best checkpoint is selected according to the DNSMOS score of the validation set.

For the sound source localization task, the window length of STFT is 512 samples (32 ms) with a frame shift of 320 samples (20 ms). The model outputs one localization result for every 5 frames. The Adam is used as the optimizer for training. The batch size of the fixed-array model and variable-array model are set to 16 and 4, respectively. The learning rate is initially set to 0.0005, and exponentially decays with a decaying factor of 0.975. We train each model for 50 epochs and the best checkpoint is selected according to the validation loss.

## D.2    Analysis of the spatial coherence of real multi-channel noise

One of the major differences between simulated diffuse noise and real-world recorded noise manifests as the spatial coherence [1]. We show the spatial correlation coefficient of the noise signals recorded with a 6cm-spaced microphone pair in different scenes, and the theoretical spherical (3D diffuse) noise field in Fig. 9. It can be seen that the spatial correlation of real noise largely varies from scene to scene. Most of the indoor and transportation scenes have a noise field that is fairly close to the 3D diffuse noise field, with several exceptions such as LivingRoom1 and OfficeLoby. In semi-outdoor and outdoor scenes, ambient noise is a complicated and dynamic combination of (partially) directional noise sources and (partially) diffuse noise sources from open space, and correspondingly their noise field generally deviate the 3D diffuse noise field.

To demonstrate the temporal stationarity in terms of the spatial correlation of real noise, in Fig. 10. we plot the curve of spatial correlation as a function of time for the 'Market' scene, where the spatial correlations are estimated from consecutive 1-second recordings. This figure shows that, even for the 'Market' scene that has an overall 3D diffuse noise field, the spatial correlation is still highly time varying and non-stationary.

## D.3    Experiments results under high-SNR conditions

### D.3.1    Benchmark experiments

In addition to the low-SNR conditions presented in Section 4.2 where the recorded speech and noise are added, in this section, we present the results for high-SNR conditions where only the recorded

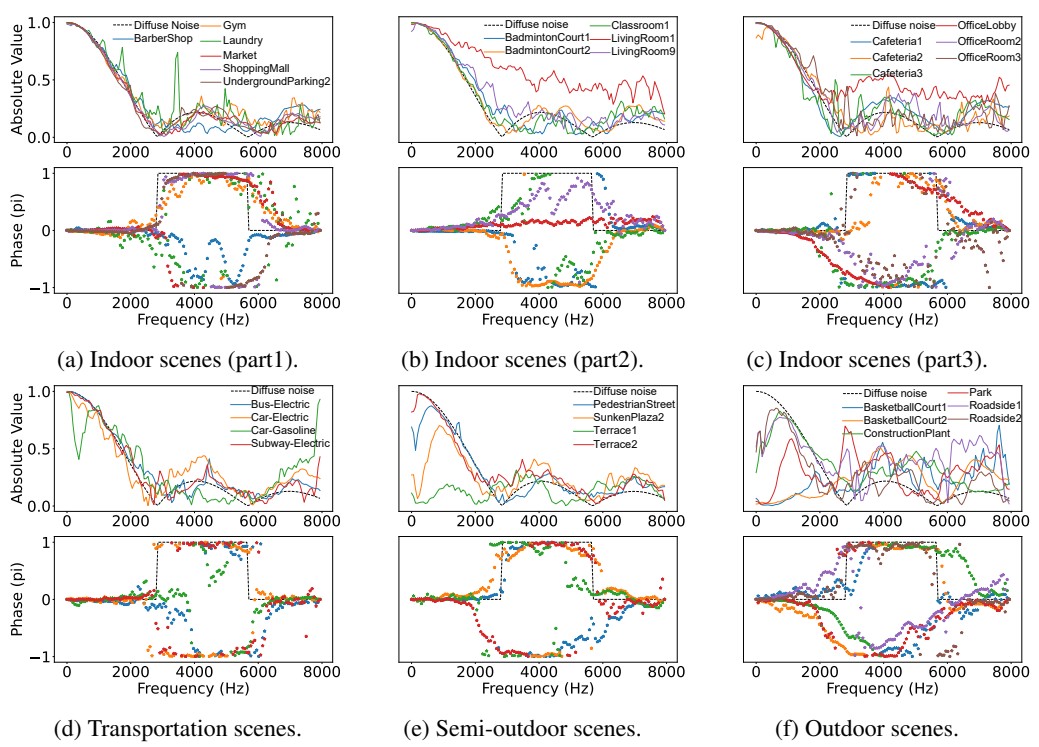

(a) Indoor scenes (part1).    (b) Indoor scenes (part2).    (c) Indoor scenes (part3).

(d) Transportation scenes.    (e) Semi-outdoor scenes.    (f) Outdoor scenes.

Figure 9: Spatial correlation coefficients of multi-channel noise in different scenes.

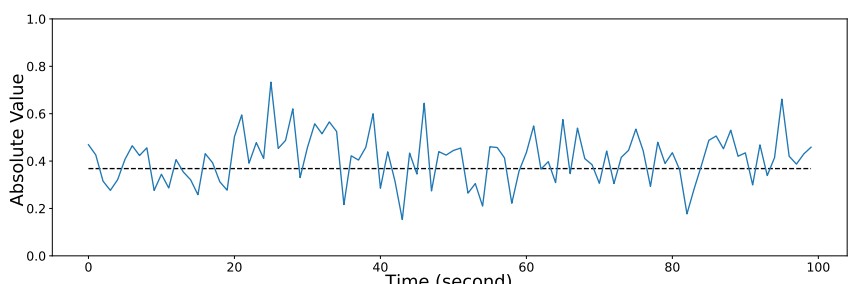

Figure 10: The (absolute value of) spatial correlation as a function of time, at the frequency of 2 kHz for the 'Market' scene.

speech (without adding extra noise) are tested. Table 10 and 11 show the performance of speech enhancement and sound source localization, respectively.

### D.3.2 Variable-array networks and array generalization

Table 12 and Table 13 show the variable-array experiments for the high-SNR case. Similar conclusions can be drawn as for the low-SNR experiments.

Table 10: Benchmark experiments of speech enhancement under high-SNR conditions.

| Baseline | Traning Data | | Static Speaker | | | | | | Moving Speaker | | | | | |
|---|---|---|---|---|---|---|---|---|---|---|---|---|---|---|
| | speech | noise | WB-PESQ | SI-SDR | MOS-SIG | MOS-BAK | MOS-OVR | CER | WB-PESQ | SI-SDR | MOS-SIG | MOS-BAK | MOS-OVR | CER |
| unprocessed | - | - | 1.34 | -4.8 | 2.59 | 2.38 | 2.00 | 9.9 | 1.20 | -4.9 | 2.28 | 2.00 | 1.70 | 10.3 |
| FaSNet-TAC [13] | sim | sim | 1.74 | -0.3 | 2.96 | 3.63 | 2.58 | 12.7 | 1.62 | -0.2 | 2.86 | 3.46 | 2.42 | 14.0 |
| | sim | real | **1.84** | 0.3 | **3.03** | 3.69 | **2.67** | **11.0** | **1.67** | 0.3 | **2.93** | 3.54 | **2.53** | **11.0** |
| | real | sim | 1.81 | **4.3** | 2.90 | **3.79** | 2.60 | 13.3 | 1.62 | **2.3** | 2.75 | **3.67** | 2.42 | 13.5 |
| | real | real | 1.83 | **4.3** | 2.97 | 3.72 | 2.63 | 12.0 | 1.64 | 2.2 | 2.87 | 3.57 | 2.49 | 11.7 |
| SpatialNet [34] | sim | sim | 2.03 | -0.9 | **3.39** | 3.48 | 2.84 | **8.9** | 1.81 | -0.8 | **3.37** | 3.34 | 2.75 | 10.0 |
| | sim | real | 1.22 | -6.0 | 1.81 | 2.09 | 1.50 | 21.6 | 1.19 | -5.8 | 1.69 | 1.90 | 1.40 | 26.2 |
| | real | sim | **2.69** | 8.0 | 3.28 | 3.68 | 2.87 | **8.9** | **2.33** | 5.2 | 3.20 | 3.55 | 2.73 | **9.9** |
| | real | real | 2.66 | **8.6** | 3.32 | **3.69** | **2.90** | **8.9** | **2.33** | 5.2 | 3.24 | **3.62** | **2.80** | 10.6 |

Table 11: CRNN benchmark experiments of sound source localization under high-SNR conditions.

| Training Data | | Static Speaker | | Moving Speaker | |
|---|---|---|---|---|---|
| speech | noise | ACC(5°) [%] | MAE [°] | ACC(5°) [%] | MAE [°] |
| sim | sim | 75.2 | 3.1 | 74.5 | 3.5 |
| sim | real | 64.8 | 10.3 | 59.7 | 9.6 |
| real | sim | **92.7** | **1.8** | 92.8 | 2.1 |
| real | real | 90.9 | **1.8** | 92.8 | **2.0** |

Table 12: FaSNet-TAC variable-array experiments for speech enhancement under high-SNR conditions.

| Setting | Static Speaker | | | | | | Moving Speaker | | | | | |
|---|---|---|---|---|---|---|---|---|---|---|---|---|
| | WB-PESQ | SI-SDR | MOS-SIG | MOS-BAK | MOS-OVR | CER | WB-PESQ | SI-SDR | MOS-SIG | MOS-BAK | MOS-OVR | CER |
| unprocessed | 1.34 | -4.8 | 2.59 | 2.38 | 2.00 | 9.9 | 1.20 | -4.9 | 2.28 | 2.00 | 1.70 | 10.8 |
| Fixed-Array | **1.77** | **4.2** | **2.94** | 3.70 | **2.59** | 13.7 | **1.59** | **1.5** | **2.83** | 3.54 | **2.43** | **15.6** |
| Variable-Array | 1.75 | 3.5 | 2.92 | **3.75** | **2.59** | 14.1 | 1.58 | 1.3 | 2.78 | **3.59** | 2.42 | 16.1 |

Table 13: IPDnet variable-array experiments for sound source localization under high-SNR conditions.

| Setting | Static Speaker | | Moving Speaker | |
|---|---|---|---|---|
| | ACC(5°) [%] | MAE [°] | ACC(5°) [%] | MAE [°] |
| Fixed-Array[15] | **89.0** | **2.1** | **91.4** | **2.1** |
| Variable-Array[15] | 88.3 | 2.3 | 84.3 | 2.9 |

