# A Supplementary materials

## A.1 Documentation and intended uses

We include a datasheet in Section B.

## A.2 Data access and code availability

We have released the RealMAN dataset on the AISHELL website to ensure its long-term preservation. Researchers can access our dataset either directly at the dataset website `https://www.aishelltech.com/RealMAN` or find the dataset website at the GitHub repository `https://github.com/Audio-WestlakeU/RealMAN`.

The detailed information and code about the RealMAN dataset can be found at the GitHub repository `https://github.com/Audio-WestlakeU/RealMAN`. This link also gives a detailed description of how to use recordings from the RealMAN dataset for speech enhancement and localization.

## A.3 Data format

RealMAN dataset provides audio files in the format of 'flac', position annotation files in the format of 'csv', and a transcription file in the plain text format of 'trn', all of which can be read by a variety of tools.

## A.4 Hosting and maintenance plan

RealMAN will remain hosted on the AISHELL website and maintained by the AudioLab of Westlake University for the foreseeable future. Any changes will be updated on the Github repository.

## A.5 Statement of responsibility

The authors declare that they bear all responsibility for violations of rights and that this dataset is released under CC-BY-4.0 license. During the recording process, we adhered to the Personal Information Protection Law of the People's Republic of China (China's PIPL) and explained all privacy-related details to the participants. We ensured the participants were fully informed and obtained the consent of all involved.

## A.6 License

The RealMAN dataset is made available under the CC BY 4.0 license. The authors bear all responsibility in case of violation of rights.

## A.7 Potential negative societal impacts

Our dataset has no potential negative societal impact.

## A.8 Personally identifiable information or offensive content

Our dataset has no personally identifiable information or offensive content.

# B Datasheet

## B.1 Motivation

**1. For what purpose was the dataset created?**

The training of deep learning-based multichannel speech enhancement and source localization systems relies heavily on the simulation of room impulse response and multichannel diffuse noise, due to the

lack of large-scale real-recorded datasets. However, the acoustic mismatch between simulated and real-world data could degrade the model performance when applying in real-world scenarios. To bridge this simulation-to-real gap, this paper presents a new relatively large-scale Real-recorded and annotated Microphone Array speech&Noise (RealMAN) dataset.

**2. Who created this dataset (e.g., which team, research group) and on behalf of which entity (e.g., company, institution, organization)**

The RealMAN dataset was created by AudioLab of Westlake University and Beijing AIShell Technology Co. Ltd.

**3. Who funded the creation of the dataset?**

This work was supported by the Zhejiang Provincial Natural Science Foundation of China under Grant 2022XHSJJ008 and the Postdoctoral Science Foundation of China under Grant 2022M722848.

**B.2   Composition**

**1. What do the instances that comprise the dataset represent (e.g., documents, photos, people, countries)?**

The instances of RealMAN are a series of files, which include real-recorded 32-channel speech and noise recordings in the format of 'flac', direct-path clean signas in the format of 'flac', and position annotation files in the format of 'csv'.

**2. How many instances are there in total (of each type, if appropriate)?**

The dataset consists of 83 hours of speech and 144 hours of noise, recorded in 32 and 31 different scenes, respectively. In addition, this dataset provides annotations of source azimuth angle, direct-path target clean speech, and speech transcription.

**3. Does the dataset contain all possible instances or is it a sample (not necessarily random) of instances from a larger set?**

The RealMAN dataset is recorded from scratch, and is not sampled from a larger set.

**4. What data does each instance consist of?**

Each instance of the RealMAN dataset consists of microphone signal recordings and the corresponding annotations.

**5. Is there a label or target associated with each instance?**

Yes. The RealMAN provides a series of labels/targets including source azimuth angle, direct-path target clean speech, and speech transcription, for speech enhancement and source localization.

**6. Is any information missing from individual instances?**

No.

**7. Are relationships between individual instances made explicit (e.g., users' movie ratings, social network links)?**

N/A

**8. Are there recommended data splits (e.g., training, development/validation, testing)?**

Yes. We split them into training, validation and testing sets according to the acoustic characteristics of the recording scenes and speaker identities. The total 83.9 hours of the recorded speech are divided into 63.5, 7.8 and 11.6 hours for training, validation and test, respectively. And 144.5 hours of noise data are divided into 106.3, 16.0 and 22.2 hours for training, validation and test, respectively.

**9. Are there any errors, sources of noise, or redundancies in the dataset?**

No.

**10. Is the dataset self-contained, or does it link to or otherwise rely on external resources (e.g., websites, tweets, other datasets)?**

The RealMAN dataset is mostly self-contained, but the speech source signals played by the loudspeaker are obtained from AISHELL.

**11. Does the dataset contain data that might be considered confidential (e.g., data that is protected by legal privilege or by doctor-patient confidentiality, data that includes the content of individuals' non-public communications)?**

No. For speech recordings, all the source speech signals are collected by AISHELL, which are free talk and reading. For noise recording, we are mostly recorded in public scenes and a voice activity detection method is used to filter out the recordings including prominent speech, so it avoids the contents of individuals' non-public communications.

**12. Does the dataset contain data that, if viewed directly, might be offensive, insulting, threatening, or might otherwise cause anxiety?**

No.

**13. Does the dataset relate to people?**

No.

## B.3    Collection process

**1. How was the data associated with each instance acquired?**

The objective of the speech recording process is to mirror real-life scenarios of human activities. In each scene, the position of both the camera and microphone array are fixed. When playing source speech, the position of the loudspeaker takes on either static or moving states. For the moving case, one person manually moves the loudspeaker carrier with varying but reasonable moving speed. In transportation scenarios, people typically maintain a stationary position, thereby the loudspeaker only takes the static state. The height of the microphone array is set to 1.40 m. The center height of the loudspeaker is aligned with the height of the mouth of a standing person, varying randomly between 1.30 m and 1.60 m. Most of the time, the loudspeaker faces towards the microphone array. We ensure that most speech recordings were conducted under quiet conditions (usually at midnight), with background noise levels maintained below 40 dB. Noise recording is simpler, for which we place the microphone array in various environments to capture the real-world ambient noise. Noise recording is normally conducted in the daytime with active events in each environment. Simultaneously, we have developed algorithms to perform target annotations for tasks related to speech enhancement and sound source localization.

**2. What mechanisms or procedures were used to collect the data (e.g., hardware apparatus or sensor, manual human curation, software program, software API)?**

A 32-channel microphone array has been proposed for audio signals recording (including noise and speech signals played through speakers). The fisheye camera captures a 360-degree panoramic image in real time, synchronized with the microphone recording. We utilize this 360-degree panoramic image for azimuth annotation. The specific introduction of the equipment is as follows:

- 32-channel microphone array is comprised of 32 high-fidelity Audio-Technica BP899 microphones. The array geometry is shown in the RealMAN paper. The audio signals are then digitized by 4 clock-synchronized 8-channel microphone pre-amplifiers (RME OctoMic II) and processed by a laptop through an audio interface (Digiface USB).

- The 360-degree fisheye camera (HIKVISION DS-2CD63C5F-IHV) is placed right above the microphone array. The frame rate of the fisheye camera is 100 ms.

- A high-fidelity monophonic loudspeaker (FOSTEX 6301 NE) is used to play source speech signals. It is placed on a height-adjustable and mobile carrier such that one can control the

position of the loudspeaker to mimic a standing/moving human speaker. A 5-cm diameter LED light is put on the top of the loudspeaker to magnify the visibility of loudspeaker to the the fisheye camera and annotate the position of the loudspeaker. The LED light can emit red or green light, which is visible for the fisheye camera under various of light conditions.

**3. If the dataset is a sample from a larger set, what was the sampling strategy (e.g., deterministic, probabilistic with specific sampling probabilities)?**

N/A

**4. Who was involved in the data collection process (e.g., students, crowdworkers, contractors) and how were they compensated (e.g., how much were crowdworkers paid)?**

We did not employ external crowdworkers or contractors for data collection.

**5. Over what timeframe was the data collected?**

Data collection starts in March 2022, and ends in May 2024.

**6. Were any ethical review processes conducted (e.g., by an institutional review board)?**

No.

**B.4 Preprocessing/cleaning/labeling**

**1. Was any preprocessing/cleaning/labeling of the data done (e.g., discretization or bucketing, tokenization, part-of-speech tagging, SIFT feature extraction, removal of instances, processing of missing values)?**

Yes. We provide target annotations for speech enhancement and sound source localization tasks, while also removing clearly identifiable speech segments from the noise signals.

**2. Was the "raw" data saved in addition to the preprocessed/cleaned/labeled data (e.g., to support unanticipated future uses)? If so, please provide a link or other access point to the "raw" data.**

No.

**3. Is the software used to preprocess/clean/label the instances available? If so, please provide a link or other access point.**

No.

**B.5 Use**

**1. Has the dataset been used for any tasks already?**

Yes. In this paper, we have used it in speech enhancement (denoise and dereverberation) and speaker localization.

**2. Is there a repository that links to any or all papers or systems that use the dataset?**

Not at the present time.

**3. What (other) tasks could the dataset be used for?**

The RealMAN dataset can be used for speech enhancement and speaker localization.

**4. Is there anything about the composition of the dataset or the way it was collected and preprocessed/cleaned/labeled that might impact future uses?**

No.

**5. Are there tasks for which the dataset should not be used?**

No.

### B.6    Distribution

**1. Will the dataset be distributed to third parties outside of the entity (e.g., company, institution, organization) on behalf of which the dataset was created?**

We have distributed the RealMAN dataset to the AISHELL website. Maybe we will distribute it to third parties.

**2. How will the dataset will be distributed (e.g., tarball on website, API, GitHub)? Does the dataset have a digital object identifier (DOI)?**

The full dataset is already publicly accessible on `https://www.aishelltech.com/RealMAN`.

**3. When will the dataset be distributed?**

It is already distributed.

**4. Will the dataset be distributed under a copyright or other intellectual property (IP) license, and/or under applicable terms of use (ToU)?**

The RealMAN dataset uses the CC BY 4.0 license.

**5. Have any third parties imposed IP-based or other restrictions on the data associated with the instances?**

No.

**6. Do any export controls or other regulatory restrictions apply to the dataset or to individual instances?**

No.

### B.7    Maintenance

**1. Who is supporting/hosting/maintaining the dataset?**

RealMAN is hosted on AISHELL website and maintained by the AudioLab of Westlake University.

**2. How can the owner/curator/manager of the dataset be contacted (e.g., email address)?**

Contact us by email or Github.

**3. Is there an erratum?**

Not at the present time. Future errata will be published on the RealMAN Github Page.

**4. Will the dataset be updated (e.g., to correct labeling errors, add new instances, delete instances)?**

We may update the dataset by adding speech recorded in more scenes to make it more valuable. Any updates will be posted updates on the RealMAN Github page.

**5. If the dataset relates to people, are there applicable limits on the retention of the data associated with the instances (e.g., were individuals in question told that their data would be retained for a fixed period of time and then deleted)**

No.

**6. Will older versions of the dataset continue to be supported/hosted/maintained?**

We expect any future changes to be additive. If that changes, we will release our versioning policy on the RealMAN Github page.

**7. If others want to extend/augment/build on/contribute to the dataset, is there a mechanism for them to do so?**

We accept feedback in the issues of the RealMAN Github page.