# OpenReview forum: "RealMAN: A Real-Recorded and Annotated Microphone Array Dataset for Dynamic Speech Enhancement and Localization"
_NeurIPS.cc/2024/Datasets_and_Benchmarks_Track — NeurIPS 2024 Track Datasets and Benchmarks Poster_

### Official Review · Reviewer_R9w9 · 2024-07-04
**An official review**

**Rating:** 5
**Confidence:** 5

**Review:**

The dataset addresses an important limitation of existing datasets, by providing data which realisticness, quantity, and diversity exceeds some of the available datasets. The data seems to be collected in a solid manner.

There are quite many important details missing from the paper. It has not been described how the exact locations of the microphone array and the speakers where chosen (only the heights are given), and in case the loudspeaker was moving, what kind of trajectories were used.  It has not been described how much data is collected from each scene, and how much content from one speaker there is.

Modeling the observed signal in terms of device-dependent impulse response, attenuation and delay is interesting. However, there are not enough details about this model and measurements related to it, for example how the impulse responses related to the devices are  obtained, and whether they are multichannel or single channel, and how the spatial properties of the devices are taken into account in the model. It is also not described what was the orientation of the loudspeaker during the data collection.

The paper estimates direct-path speech signals by using a procedure that measures the delay and attenuation between source speech signals and measured signals. The procedure presented in Algorithms 1 and 2 contains many steps that are quite vaguely presented (for example "discarding the unreliable data", "discard the unreliable speech pairs", etc.). Therefore it is not fully clear how the reference direct-path speech signals are produced. It would have been great to have a more thorough evaluation to show that the procedure leads to good estimates. Now the paper only refers to some ASR experiments where the generated direct path speech signals give the same CER than original source speech. However, that only measures the intelligibility of ¸generated signals, measuring their other properties would have added the credibility of the procedure.

In the evaluation, it is not clear how the simulated data is generated. How speech and noise segments to be mixed are chosen, and what are the RIRs used? I don't even see it being described what is the simulated noise or RIRs that are used.

The paper also refers to some mismatches between real and designed microphone positions, which has apparently had a huge affect on the results. This issue should have been discussed more thoroughly, to understand the implications on the dataset.

In the localization experiments, the sentence "both real speech and real noise are beneficial for improving the sound source localization perforemance", it is not clear in comparison to what the performance improves.

**Strengths:**

The dataset would be important for speech enhancement and localization research, and its realisticness, diversity, and quantity partially exceeds many existing datasets.

**Additional Feedback:**

This would be a great dataset if the missing information that I have mentioned above will be added.

**Clarity:**

The paper is in general quite well written. Many important details as missing (see the above review). There are some minor problems with the English language, and the paper would benefit from proof reading.

**Correctness:**

It seems that the dataset has been constructed in a sound way, but there is lots of missing information so that not all the aspects cannot be properly evaluated.

**Documentation:**

Lots of details about the data collection procedure are missing.

**Ethics:**

I do not see any ethical issues.

**Limitations:**

Some limitations have been observed in the paper, but much more though treatment would have been credible. See the review for details of missing information.

**Opportunities For Improvement:**

Lots of important information is missing from the paper.

**Relation To Prior Work:**

The paper discusses and summarizes properties of existing datasets very well. The paragraph that compares the proposed dataset to the previous ones in the end of section 2 is somewhat vague. It does not properly discuss how the properties of the collected dataset specifically differ from the existing datasets.

**Summary And Contributions:**

The paper proposes a dataset consisting of multichannel recordings of speech played by a loudspeaker captured in diverse set of various locations, as well as noise recorded in these locations. The intended use of the dataset is speech enhancement and localization research. The paper also presents baseline experiments on these topics with the dataset, including also simulated data to evaluate the usefulness of the real data.

---

> ### Author Response · Authors · 2024-08-17
>
> We are grateful for your valuable comments and detailed advises on our manuscript.
>
> We are currently finalizing our rebuttal materials in response to your concerns.
>
> The complete rebuttal materials would be uploaded by 11:59PM August 17th, AOE.
>
> We apologize for any inconvenience caused by the delay in meeting the original deadline.
>
> Authors

---

> > ### Author Rebuttal · Authors · 2024-08-18
> >
> > ---
> > - "However, that only measures the intelligibility of ¸generated signals, measuring their other properties would have added the credibility of the procedure."
> >
> > **Response**: This is an really important question, and one major difficulty appeared in our method, as we don’t have the ground truth. We have made lots of efforts to testify the quality and accuracy of generated signals, including performing listening test, ASR, observing/improving the credibility of cross-correlation. Most importantly, the generated signals can be used for successfully training the speech enhancement network. This issue will be more clearly presented in the revised manuscript as:
> >
> > "Due to the lack of ground truth, it is not straightforward to evaluate the estimation accuracy. We think the credibility of the estimated direct-path speech can be well testified based on the following criteria. i) The estimated direct-path speech should have the same speech quality as source speech to provide an ideal upper-bound for speech enhancement. We evaluate the speech quality with two metrics. One metric is the subjective speech quality. We have conducted an informal listening test by eight listeners to test whether there is an audible difference between the perceptual quality of the estimated speech and source speech, which shows that there is no audible difference with a 91% confidence. The second metric is the ASR performance, tested with an established ASR model trained on over 10,000 hours of Mandarin dataset WenetSpeech using the ESPNet toolkit. The character error rates (CERs) (and thus the speech intelligibility) for the estimated direct-path speech and source speech are identical. ii) As long as speech enhancement networks can be successfully trained with the estimated direct-path speech as target (will be shown in the section of experiments), the estimated direct-path speech can be deemed to be sufficiently accurate. iii) As shown in Appendix, the estimation of $A$ and $\tau$ is based on the cross-correlation method. The good properties of intermediate results, such as the sharp correlation peak and the smooth estimation curve for moving source, also give us strong confidence on the estimation accuracy. "
> >
> > ---
> > - "In the evaluation, it is not clear how the simulated data is generated."
> >
> > **Response**: Some incomplete information about data simulation have been presented in the manuscript, and we will further clarify them in the revised manuscript as:
> >
> > “Equal amounts of multichannel speech are simulated according to our real-recorded dataset, using the gpuRIR toolkit [gpuRIR]. Specifically, one counterpart utterance is simulated for each real-recorded utterance using the same source speech, room size, T60, and source position/trajectory as the real-recorded utterance. Multi-channel noise is simulated with the diffuse noise generator [DiffuseNoise08], taking white (code generated), babble and factory noise (from the noisex-92 dataset [noisex-92]) as the single-channel source noise.”
> >
> > ---
> > - "The paper also refers to some mismatches between real and designed microphone positions, which has apparently had a huge affect on the results. This issue should have been discussed more thoroughly, to understand the implications on the dataset."
> >
> > **Response**: we have conducted a large amount of validation experiments on this issue, even after the paper submission. Now we think the microphone position mismatch does exist, but it is not the main reason of simulation-to-real mismatch. The main reason is still unclear, which we think it is reasonable as there exist many unclear simulation-to-real factors, and analyzing/reducing the mismatch is a research topic by itself.
> >
> > This issue will more thoroughly presented in the revised manuscript as:
> >
> > “Some validation experiments had been conducted to figure out the reasons for this phenomenon, which showed that there might be a slight mismatch between the real and ideal microphone positions. We have designed a new method for resolving the problem of microphone position mismatch, namely disturbing the ideal microphone positions in training, which is shown to be very effective on simulated test data, but only slightly improve the performance on our real test data. This indicates that there are still some unclear mismatches between the simulated and real data. As its name indicates, SpatialNet mainly learns the RIR-related spatial information, which is possibly more sensitive to those unclear mismatches. Resolving those mismatches would be an interesting topic for future research.”
> >
> > ---
> > - "In the localization experiments, the sentence "both real speech and real noise are beneficial for improving the sound source localization perforemance", it is not clear in comparison to what the performance improves."
> >
> > **Response**: We think, in Table 5, all the performance metrics consistently show that using real speech/noise outperforms using simulated speech/noise.

---

> ### Author Rebuttal · Authors · 2024-08-18
>
> Thank you very much for your insightful and valuable comments, which are very helpful for us to further revise the manuscript.
>
> We apologize for missing the important information you have mentioned. We can add all of them in the revised manuscript. Details are as follows.
>
> ---
>
> - "There are quite many important details missing from the paper. It has not been described how the exact locations of the microphone array and the speakers where chosen (only the heights are given), and in case the loudspeaker was moving, what kind of trajectories were used. "
>
> **Response**:
>
> 1. The microphone array location will be more clearly presented in the revised manuscript as:
>
> “In each scene, the microphone array is located at one position around the scene center part, and sometimes it is put close (not very close) to one side wall for having a large source-to-array distance or not disturbing other people.”
>
> 2. The speaker state (movement and orientation) will be more clearly presented in the revised manuscript as:
>
> “For static speaker, most of the time, the loudspeaker faces towards the microphone array, with a small portion of side-facing (but no back-facing). For moving speaker, movements include a large portion of large-range walking, a small portion of small-range pacing and a smaller portion of head turning. The moving trajectory of large-range walking can be axial, tangential or in between, and can be random walking as well. During movement, the loudspeaker orientation is put either facing (most of the time side-facing) the microphone array, or towards the moving direction. The source-to-array distances are mainly distributed in the range of 0.5 m - 5 m. Overall, in the proposed dataset, a number of speaker states are considered. However, we think it is still far from exhausting the speaker states in various speech applications, and the influence of speaker states on speech enhancement and source localization would be an interesting topic for future research.”
>
> 3. Before, we only provide the azimuth annotation, but now we can provide the full annotation of source location including azimuth, elevation and distance. The statistics of source location (azimuth, elevation and distance) and several examples of moving trajectory will be added, please see the attached extra pdf file.
>
> ---
> - "It has not been described how much data is collected from each scene, and how much content from one speaker there is."
>
> **Response**: This will be added, please see Table I in the attached extra pdf file.
>
> ---
> - "However, there are not enough details about this model and measurements related to it, for example how the impulse responses related to the devices are obtained, and whether they are multichannel or single channel, and how the spatial properties of the devices are taken into account in the model."
>
> **Response**: We will clarify related information in the revised manuscript as:
>
> “The impulse response of devices (microphone and loudspeaker combined) $h_{dev}$ is considered to be constant (independent to the direction and orientation of the loudspeaker relative to the microphone) and it is measured in advance for one configuration where the loudspeaker faces toward the microphone with a source-to-microphone distance of 1 m, please refer to Appendix~measurement for more details. This simplified measurement is accurate for the omni-directional microphone (except for a constant time delay and level factor), but inaccurate for the frontal-directional loudspeaker as its impulse response is orientation-dependent. However, in our setting, it is difficult to annotate the loudspeaker orientation and thus to take the loudspeaker directivity into account, which is left for future research.
> Any constant time delay and level factor, such as the one caused by the measurement of $h_{dev}$ or by the loudspeaker orientation, will be absorbed into the estimated $\tau$ and $A$, respectively. The direct-path target speech can be estimated in the same way for all microphone channels. As we want to keep a small data size, only the direct-path target speech for microphone 0 is included in the released dataset. In the future, we can provide more if there is a high requirement.”
>
> ---

---

> > ### Comment · Reviewer_R9w9 · 2024-08-28
> >
> > Thank you for the additional information, this clearly improves the quality of the manuscript. I am happy to update my evaluation.

---

> > > ### Author Rebuttal · Authors · 2024-08-30
> > >
> > > Dear Reviewer,
> > >
> > > Thank you once again for your valuable comments and for your feedback to our rebuttal. However, we noticed that you still rated our work as "marginally below the acceptance threshold".
> > >
> > >
> > > We think that you may recognize the quality of our dataset, as you mentioned that "This would be a great dataset if the missing information I have mentioned above is added." So, we would like to know if you believe there are any other missing information or drawbacks in our work.
> > >
> > >
> > > If so, please let us know, as we are willing to further improve the quality of our work.
> > >
> > > Thank you,
> > >
> > > Authors

---

### Official Review · Reviewer_UpSw · 2024-07-18
**A microphone array dataset for speech enhancement and localization**

**Rating:** 7
**Confidence:** 4
**Correctness:** The authors correctly claims the data…
**Clarity:** This paper is easy to follow.

**Review:**

See the below.

**Strengths:**

Their real recording for moving sources avoids issues associated with the piece-wise generation method.
They also serve the direct-path speech as the training target signal for speech enhancement using a filtering method.
These help benchmark for enhancement and localization tasks.

The number of scenes are over 30, and duration of recording are 83 hours.
The long recordings with 32-ch microphone array helps reliable evaluation.

**Additional Feedback:**

Let me ask about the experiment on Table 6.
In IPDnet paper [12], the authors say "the DOA estimation can be obtained by simply matching the predicted DP-IPDs with the DP-IPD templates".
In the experiment on Table 6, is the DP-IPD templates also used?
Are the templates calculated from the array geometry in the same manner of [12]?

The authors could consider the below microphone array datasets for source localization.
While the datasets focus on various sound events in addition to speech, they also serve microphone array recordings and location information.
* "SECL-UMons Database for Sound Event Classification and Localization," Brousmiche+, ICASSP 2020.
* "STARSS23: An Audio-Visual Dataset of Spatial Recordings of Real Scenes with Spatiotemporal Annotations of Sound Events," Shimada+, NeurIPS DBT 2023.

**Documentation:**

The authors provides sufficient detail regarding the data collection process, organization, and availability.

**Ethics:**

There is no ethical concern of this submission.

**Limitations:**

See the Opportunities For Improvement.

**Opportunities For Improvement:**

As they mentioned in section 2, an unreal point is that they use a loudspeaker playing back speech, instead of speaking by real human speakers.
While the authors move the loudspeaker to mimic the human walking, the movement of human speakers are not perfectly reproduced.
The content and style of speech could change to some extent.

**Relation To Prior Work:**

It is discussed how this work is different from previous contributions.

**Summary And Contributions:**

The authors present a microphone array dataset for speech enhancement and localization.
A 32-channel microphone array is used for recording, and a loudspeaker is used for playing source signals.
A total of 83-hour static or moving speech signals are recorded in 32 scenes, and 144 hours of background noise are recorded in 31 scenes.
The direct-path signal, the target for speech enhancement, is obtained by filtering the source signal with an estimated direct-path propagation filter.

---

> ### Author Response · Authors · 2024-08-17
>
> We are grateful for your valuable comments and detailed advises on our manuscript.
>
> We are currently finalizing our rebuttal materials in response to your concerns.
>
> The complete rebuttal materials would be uploaded by 11:59PM August 17th, AOE.
>
> We apologize for any inconvenience caused by the delay in meeting the original deadline.
>
> Authors

---

> ### Author Rebuttal · Authors · 2024-08-18
>
> Thank you very much for your valuable comments, which are very helpful for us to further revise the manuscript.
>
> "As they mentioned in section 2, an unreal point is that they use a loudspeaker playing back speech, instead of speaking by real human speakers. While the authors move the loudspeaker to mimic the human walking, the movement of human speakers are not perfectly reproduced. "
>
> **Response**: Yes, using a loudspeaker indeed brings some unrealistic factors. However, recording with human speakers will significantly increase the cost for both recording and annotation, and will certainly constrain the recording quantity.
>
> "Let me ask about the experiment on Table 6. In IPDnet paper [12], the authors say "the DOA estimation can be obtained by simply matching the predicted DP-IPDs with the DP-IPD templates". In the experiment on Table 6, is the DP-IPD templates also used? Are the templates calculated from the array geometry in the same manner of [12]?"
>
> **Response**: Yes, the DP-IPD templates are also used, and they are calculated from the array geometry in the same manner of [12]. This information will be clarified in the revised paper.
>
> "The authors could consider the below microphone array datasets for source localization. While the datasets focus on various sound events in addition to speech, they also serve microphone array recordings and location information."
>
> **Response**: Yes, the datasets you mentioned are great and popular. We will add the reference and compare to them. Our proposed dataset provides a larger data size than for example the STARSS23 dataset, i.e. 83 hours versus 10.9 hours.

---

> > ### Comment · Reviewer_UpSw · 2024-08-30
> > **Response to rebuttal**
> >
> > Thank you for your clarification of DP-IPD and addition of existing datasets. I have modified my review.

---

### Official Review · Reviewer_Uhcz · 2024-07-25
**This paper introduced a dataset for real-recorded microphone array data for speech.**

**Rating:** 6
**Confidence:** 2
**Correctness:** Dataset is constructed in a sound way.
**Clarity:** There are points that should be elabo…

**Review:**

The proposed dataset may improve robustness of speech-related tasks, such as ASR in far-field scenarios. Experiments show that the dataset can reduce the simulation-to-reality gap for speech enhancement and localization. The use of 32-array microphones may be a robust contribution compared to prior datasets.

**Strengths:**

- Method for collection speech data is clearly described, and the geometry of the array aptly described.
- A variety of outdoor, and indoor scenes are collected, representing a diverse set of scenarios for speech applications.
- Number of channels is larger than existing datasets, facilitating the training of variable-array neural nets.

**Additional Feedback:**

-

**Documentation:**

Documentations is sufficient.

**Ethics:**

No.

**Limitations:**

Authors adequately discussed the limitations.

**Opportunities For Improvement:**

- Dataset is not described in detail:
  - Until section 3.2, it is not mentioned that the dataset contains speech in Mandarin Chinese.
  - What kind / topic of speech does it contain?
  - What language register / formality does the speech use?
  - Does the dataset provide gold transcriptions to accompany the speech data?
- Table 1: what is the unit of the column in `# RIR / speech duration`?

**Relation To Prior Work:**

Relation to prior work is clearly discussed.

**Summary And Contributions:**

The paper introduced a dataset of real-recorded microphone 32-channel array data, with both audio and noise. It is used for benchmarking speech enhancement and localization methods. The dataset is recorded in a real room instead of simulated in a empty-boxed shaped theoretical room.

---

> ### Author Response · Authors · 2024-08-17
>
> We are grateful for your valuable comments and detailed advises on our manuscript.
>
> We are currently finalizing our rebuttal materials in response to your concerns.
>
> The complete rebuttal materials would be uploaded by 11:59PM August 17th, AOE.
>
> We apologize for any inconvenience caused by the delay in meeting the original deadline.
>
> Authors

---

> ### Author Rebuttal · Authors · 2024-08-18
>
> Thank you very much for your valuable comments, which are very helpful for us to further revise the manuscript.
>
> "Until section 3.2, it is not mentioned that the dataset contains speech in Mandarin Chinese."
>
> **Response**: we will clarify this information in the abstract and introduction parts of the paper.
>
> "What kind / topic of speech does it contain?"
>
> **Response**: For free talk, speakers are encouraged to converse alone. Reading speech entail speakers reading news articles. The topics of speech content spread a wide range of domains including news reports, games, reading experiences, and life trivia.
> This will be further clarified.
>
> "Does the dataset provide gold transcriptions to accompany the speech data?"
>
> **Response**: Yes, the dataset provides the transcriptions.
>
> "Table 1: what is the unit of the column in # RIR / speech duration?"
>
> **Response**: The unit for RIR is number, and hour for speech duration. This will be further clarified.

---

> > ### Comment · Reviewer_Uhcz · 2024-08-28
> >
> > Thanks for the response -- they have addressed my concerns. I have modified my scores accordingly.

---

### Official Review · Reviewer_DaAx · 2024-07-25
**Useful addition in n area of datasets that is currently lacking**

**Rating:** 7
**Confidence:** 5
**Correctness:** The work seems correct, as far as a d…

**Review:**

The work is clearly written and presented.

In terms of originality and significance, as with most dataset papers these qualities are limited. The value of the work comes form the fact that there is a clear need for such data at the moment and the authors have put a substantial effort to collect a large amount of them to address that in a comprehensive manner. The dataset is also not conclusive - it is only one out of a hundred more datasets like that which will be needed to be published for comprehensive real-world training and testing of speech enhancement, speaker localization, or separation, but it is a strong early contribution.

The work is also not perfect - it falls short in a number of aspects in terms of realism, but these shortcomings are not strong enough to outweigh the benefits that it brings, because, as mentioned above, there are very few datasets in that space, often smaller and with similar or stronger shortcomings.
More specifically:
- The authors are using a loudspeaker as the speech source, instead of recording real speakers. Of course that simplifies the annotation procedure vastly, especially in terms of spatial locations, but it also removes some realistic conditions that have an effect, or their effect should be assessed, in all the aforementioned speech processing tasks.
- The loudspeaker has a fixed directivity across all recordings - the authors do not present information on it and how close it is to average speech directivities across genders available in literature. Natural variations of directivities related to age, gender, etc.. are not included in the recordings.
- Even more importantly, the directivity effect is factored out of the recordings even for the loudspeaker, since they are done with the loudspeaker facing the microphone array at all times. This is of course an unrealistic scenario; people do not face the recording device when talking, they face each other. Having the loudspeaker facing the array is a best case scenario since it maximises the energy of the direct path signal reaching the array with respect to the reverberant energy. In realistic scenarios it would be common that a speaker facing away from the array would result in negative (in dB) such direct-to-reverberant energy ratios.
- Motion is done in moving trajectories with the loudspeaker facing the array. Even though large motions such as those are interesting evaluation material for the speaker localization tasks - they are of course less common that natural speech scenarios, where people are often stay around the same spot when addressing others, with small translation but often large rotational movements (hence related again to stronger directivity effects). Such material is not included in the recordings.
- The fact that a loudspeaker was used also takes away the originality of about half the dataset of static material, since it is practically equivalent to synthesising speech with measured loudspeaker RIRs in rooms, of which there is a wealth of datasets at the moment. However, emulated reverberant speech using real RIR datasets can not be easily coupled with real ambient noise recordings at the moment, due to a lack of RIR datasets together with ambient noise recordings for the same consistent array configurations (with a few exceptions such as TAU-SRIR-dB)

Otherwise, the dataset has a higher amount of speech and ambience recordings than earlier ones, for a larger range of room environments, with a high number of channels that offer flexible possibilities to accommodate various array configurations. The evaluation through some basic speech enhancement and localization tasks is straightforward and clear - if not always conclusive on which aspects of those systems the real recordings bring clear or strong benefits over simulations. One proxy task for evaluation that could promote the usefulness of the dataset even further is that of multi-speaker separation.

Some strongly related works are omitted in the literature review (pointed later), but generally the literature review is of good quality and fairly complete.

Observations and some investigations into the spatial correlation of real ambience are interesting and topical.

Some specific comments:
35: "The image method ... and built-in obstacles." - inaccurate - the ISM can simulate any geometry, and it serves the basis for beam tracing and other more advanced geometrical acoustics methods. However, ISM is very fast only for shoebox geometries - important for generating diverse high-volume acoustic simulations for training.

37: "The directivity of the sound sources and microphones... " - ISM can accomodate easily measured source and receiver directivities, and it is often done in practice.

45: "Relevant studies... real-world scenarios ." - A relevant study that studies this for sound source localization specifically, and matching ISM simulations to real world test conditions is:
- Srivastava, P., Deleforge, A., Politis, A., & Vincent, E. (2022). How to (virtually) train your sound source localizer. arXiv preprint arXiv:2211.16958.

82: "The scenario becomes even more complex with a moving source ... substantially more time-consuming." - This is inaccurate - the DCASE publication states that a single slowly moving loudspeaker emitting noise was used for the RIR measurements, which were extracted in post processing from that single excitation sequence. However, the process still sounds time-consuming.

96: "In DCASE, the RIR ..., which is very time-consuming." - same as earlier comment

138: "under various of light conditions." - wrong syntax, consider rewriting

159: "We ensure that most speech recordings ...with background noise levels maintained below 40 dB." - What was the average speech to noise ratio across scenes (or the range of it)?

161: Any information on the noise recordings? Were you logging SPLs? Could some information on those be found?

183: h_dev seems to be considered time-invariant, however it is obviously time-variant (or direction-of-radiation/direction-of-arrival dependent) for a moving source. We assume that you consider your microphones to have ideal omnidirectional responses, but the loudspeaker is directional. What are your further assumptions that allow you to do that simplification? Please include those in the text.

190: "the direct-path filtering of source speech do not degrade" - does not

194: We understand that the simpler azimuth only estimation has a wide range of practical applications without involvement of full azimuth-elevation localization. But please state that clearly since this is a limitation of the dataset compared to some of the earlier ones.

203: It is unclear how and if multiple recordings occur in the same scene. That relates to the question whether the dataset can be used for  multi-speaker separation by adding single speech recordings taken inside the same space. Please clarify. Also in case that is true - please comment on any potential problems with the speech recording background noise being summed up from multiple samples.

212: "Note that, although some scenes may overlap across sets, there is no data sample overlap among them." - Unclear, how much and why some scenes overlap across sets? Consider clarifying better.

215: "no speaker appears" - the same speaker does not appear

266: "Equal amounts of multichannel speech are simulated according to our real-recorded dataset. " - which simulation method? Shoebox ISM? Which package (if not in-house)? Provide some details on the simulator.

268: "Note that source speech, room size, and T60 of each real-recorded utterance are available in our dataset. " - Comment on whether loudspeaker directivity was included in the simulation (or matched in some way to it). If not, your simulations have more adverse reverberant conditions than the real recordings, because direct-to-reverberant energy ratios would be lower for omnidirectional sources.

273: "using single-channel of white, babble, and factory noise." - bad syntax, consider rewriting

280: "the target clean speech provided in this dataset are " - is

281: "which may leads to " - lead to

288: "between the real and designing microphone positions", and ideal simulated microphone positions

306: "Different from the speech enhancement ...... which is consistent with the findings in [40]"
Some more relevant studies that investigate this:
- Srivastava, P., Deleforge, A., Politis, A., & Vincent, E. (2023, August). How to (Virtually) Train Your Speaker Localizer. In INTERSPEECH
2023.
- Neri, M., Politis, A., Krause, D., Carli, M., & Virtanen, T. (2024). Speaker Distance Estimation in Enclosures from Single-Channel Audio. IEEE/ACM Transactions on Audio, Speech, and Language Processing.

314: all the 28 microphone data

Please pass the appendix material through a grammar and spell checker - the quality of writing is significantly worse than the main text.

**Strengths:**

- High volume and diversity in the dataset's acoustic conditions
- Extensive ambient noise recordings
- Moving speaker recordings was missing from most earlier datasets
- High channel count that can accommodate new research in variable or array-agnostic processing
- Coupling between speech recordings and ambient noise recordings in terms of array topology

**Additional Feedback:**

Nothing to add here.

**Clarity:**

It is clearly written. Small typos and bad grammar points are indicated in the main review.

**Documentation:**

I cannot see a hosting, licensing, and maintenance plan. Otherwise, I believe that these points are addressed..

**Ethics:**

I do not think so.

Public speech parts in the ambience noise recordings seem to have been removed.

**Limitations:**

Based on the main review text:
- use of loudspeaker emitting speech instead of real speakers
- no directivity effects included
- azimuth only tagging of source position

**Opportunities For Improvement:**

Based on the main review text:
- recordings of real speakers with position and head orientation tracking
- full localization tagging (azimuth, elevation, distance)
- some assumptions seem to not be explained adequately in the main text

**Relation To Prior Work:**

The work misses some important references. The most important related one is STARSS22/STARSS23 dataset papers which include recordings of moving humans in real rooms with full position annotation, talking among other action sounds. The dataset does not have only speech sounds annotated temporally and spatially, but speech constitutes the main activity in the recordings compared to the other classes of sounds.

Two more strongly related references are the papers on the DIRHA datasets and the VoiceHome-2 datasets, which include spatial annotations of real static speakers.

These are the only datasets we are aware of that have real speaker recordings with spatial annotations of them.

**Summary And Contributions:**

The contribution of the work is straightforward: There is a lack of speech recordings captured in real rooms with spatiotemporal annotations and reference clean speech recordings to be used for training and evaluation of the hundreds to thousands of works during the last few years on multichannel speech enhancement, dereverberation, speaker separation and speaker localization. The dataset contributes a substantial amount of multichannel recordings in this direction in a variety of conditions with spatial annotations.

The outcome of this contribution is that it allows better understanding of how methods will operate in realistic conditions.

A secondary contribution is that the almost half of the dataset contains dynamic scene recordings with spatial annotations, which is also lacking from available datasets at the moment.

---

> ### Author Response · Authors · 2024-08-17
>
> We are grateful for your valuable comments and detailed advises on our manuscript.
>
> We are currently finalizing our rebuttal materials in response to your concerns.
>
> The complete rebuttal materials would be uploaded by 11:59PM August 17th, AOE.
>
> We apologize for any inconvenience caused by the delay in meeting the original deadline.
>
> Authors

---

> > ### Author Rebuttal · Authors · 2024-08-18
> >
> > 6. You have raised the issue “h_dev seems to be considered time-invariant, however it is obviously time-variant”.
> >
> > **Response**: We apologize for not presenting the detailed information about h_dev, and it is indeed time-variant and is an inaccurate aspect of the proposed dataset.
> >
> > In the revised paper, we will clarify this issue as:
> >
> > "The impulse response of devices h_dev is considered to be constant (independent to the direction and orientation of the loudspeaker relative to the microphone) and it is measured in advance for one configuration where the loudspeaker faces toward the microphone with a source-to-microphone distance of 1 m. This simplified measurement is accurate for the omni-directional microphone (except for a constant time delay and level factor), but inaccurate for the frontal-directional loudspeaker as its impulse response is orientation-dependent. However, in our setting, it is difficult to annotate the loudspeaker orientation and thus to take the loudspeaker directivity into account, which is left for future research. "
> >
> > 7. You have mentioned “whether the dataset can be used for multi-speaker separation by adding single speech recordings taken inside the same space”.
> >
> > **Response**: We will clarify this issue in the revised paper as:
> >
> > “In this paper, we only perform single-speaker speech enhancement (denoising and dereverberation) and source localization. Nevertheless, we think the proposed dataset can also be used for the tasks of multi-speaker separation and localization. Multi-speaker signals can be generated by simply mixing the single-speaker signals recorded in the same scene, which will be highly consistent to the simultaneous recording of multiple speakers. One unreal factor is that the background noise in speech recordings will also be mixed, which we think is not problematic, as the SPL of background noise is low compared to the SPL of speech, i.e. 57 dB (36 dBA) versus 68 dB (61 dBA).”
> >
> > 8. You have mentioned “how much and why some scenes overlap across sets?”
> >
> > **Response**: We will clarify this issue in the revised paper as:
> >
> > “We think the scene diversity of training and test sets are both critical for the full evaluation of general-purpose speech enhancement and source localization methods, so we have to make some scene overlaps across sets, but note that there is no data sample overlap among them.”
> >
> > 9. You have posed questions about the simulation method used in the experiment.
> >
> > **Response**: we used the gpuRIR toolbox, in which the loudspeaker directivity was not included. As we cannot annotate the speaker orientation in the current setting, it seems difficult to include the loudspeaker orientation into the simulation.
> >
> > 10. You have mentioned some typos in the paper, and the bad quality of appendix material.
> >
> > **Response**: Thank you for pointing out these. We will correct the typos and carefully pass through the appendix material.

---

> > > ### Comment · Reviewer_DaAx · 2024-08-28
> > > **Response to rebuttal**
> > >
> > > Thank you for the additional information and revisions. The new clarifications improve the quality of the paper and highlight better the strengths and limitations of the dataset. The new information, levels, elevation, distance, RT60 and room information, and statistics also make the paper and the dataset more useful.
> > >
> > > Overally, my decision was positive on the dataset and report even before the rebuttal, and remains so.

---

> ### Author Rebuttal · Authors · 2024-08-18
>
> Thank you very much for your insightful and valuable comments, which are very helpful for us to further revise the manuscript.
>
> 1. One of your major concerns is about the use of loudspeaker and the directivity of loudspeaker.
>
> **Response**: Yes, using a loudspeaker is indeed a limitation of the proposed dataset.
>
> In the revised manuscript:
>
> “One important unreal factor is that we use a loudspeaker playing back speech signals, instead of speaking by real human speakers. Although the used loudspeaker has a similar frontal (on-axis) directivity as human speakers [DirPat18], one fixed loudspeaker cannot account for the various directivities across different human speakers.”
>
> 2. Also related to the directivity of loudspeaker, you mentioned that “ the loudspeaker facing the microphone array at all times” is a big problem, as it is an unrealistic scenario.
>
> **Response**: We apologize for that we simply described the speaker state as “most of the time, the loudspeaker faces towards the microphone array”, which is incomplete and misleading. We think “facing the array” has important applications, such as human-robot interaction and smart loudspeaker. Although not as much as the “facing the array” case, we indeed have some other states, such as side-facing the array, head turning, pacing, etc.
>
> The information of speaker state will be elaborated and clarified in the revised manuscript as:
>
> “In different speech applications, speakers could have very different states in terms of facing or not facing the device, static or moving, small-range pacing or large-range walking, and head turning. For example, as for human-robot interaction and smart loudspeaker, the speaker normally faces the device and could have some movements. In a meeting application, the speakers normally face each other and have head turning. During speech recording, we do consider these factors to a certain extent. For static speaker, most of the time, the loudspeaker faces towards the microphone array, with a small portion of side-facing (but no back-facing). For moving speaker, movements include a large portion of large-range walking, a small portion of small-range pacing and a smaller portion of head turning. The moving trajectory of large-range walking can be axial, tangential or in between, and can be random walking as well. During movement, the loudspeaker orientation is put either facing (most of the time side-facing) the microphone array, or towards the moving direction. The source-to-array distances are mainly distributed in the range of 0.5 m - 5 m. Overall, in the proposed dataset, a number of speaker states are considered. However, we think it is still far from exhausting the speaker states in various speech applications, and the influence of speaker states on speech enhancement and source localization would be an interesting topic for future research.”
>
> 3. One important shortcoming of the proposed dataset is that only the azimuth angle of source is annotated.
>
> **Response**: Before, we annotated source location using an inaccurate camera calibration function provided by the device provider, leading to the distance annotation for distant cases being suspect. Recently, we have resolved this problem by re-calibrate the camera, and now we can provide the full annotation of source location, including azimuth, elevation and distance. This information (and the statistics of source location) will be added in the revised paper. Please see the attached pdf file for the statistics.
>
> For your reference, the annotation of source location is conducted as:
>
> “the source location can be calculated in three steps. i) Based on camera calibration, the azimuth angle (relative to both the microphone array and camera) and elevation angle (relative to only the camera) of source can be calculated with the pixel position of LED light. ii) With the height (manually logged) and elevation angle (relative to the camera) of source, the horizontal distance between the source and camera/array can be calculated. iii) The elevation angle (relative to the microphone array) of source can be calculated using the height of microphone array, the height and horizontal distance of source.”
>
> 4. You have mentioned some inaccurate statements in the paper, about the ISM method, the RIR measurement in DCASE. You have also mentioned that several important related works are missing.
>
> **Response**: Thank you for pointing out these. We have checked that they are indeed highly relevant, and we will correct the inaccurate statements and add the missing references.
>
> 5. You have posed the question about the sound pressure level (SPL).
>
> **Response**: We have logged the SPLs during recording for each scene. Over all scenes, the SPL of speech recordings and silent backgrounds are averagely about 68 dB (61 dBA) and 57 dB (36 dBA), respectively. The SPL of noise recordings is averagely about 71 dB (58 dBA). This information and the SPLs for each scene will be added in the revised paper. Please see Table I in the attached pdf file.

---

### Author Response · Authors · 2024-08-28
**Enquiry about the rebuttals**

Dear Area Chairs,

We haven't received any feedback on our submitted rebuttals, so we wanted to verify that the rebuttals were submitted correctly and in the proper format.

Please let us know if everything is in order.

Thank you in advance for your assistance.

Best regards,

Authors

---

> ### Comment · Area_Chair_c2xT · 2024-08-28
> **Rebuttal**
>
> Dear Authors,
>
> Reminders have been sent to the reviewers.
>
> The AC.

---

### Decision · Program_Chairs · 2024-09-26

**Decision:**

Accept (Poster)

**Comment:**

This article offers a valuable dataset to the speech community as it will help with real-world use cases. Reviewers were unanimously agreeing in the relevance and quality of this paper. The rebuttal discussion even improved a few lacking points.